# `MatsubaraFunctions.jl`: An equilibrium Green's function library in the Julia programming language

Dominik Kiese[1*], Anxiang Ge[2], Nepomuk Ritz[2], Jan von Delft[2], Nils Wentzell[1]

**1** Center for Computational Quantum Physics, Flatiron Institute, 162 5th Avenue, New York, NY 10010, USA
**2** Arnold Sommerfeld Center for Theoretical Physics, Center for NanoScience, and Munich Center for Quantum Science and Technology, Ludwig-Maximilians-Universität München, 80333 Munich, Germany
*dkiese@flatironinstitute.org

November 29, 2023

## Abstract

**The Matsubara Green's function formalism stands as a powerful technique for computing the thermodynamic characteristics of interacting quantum many-particle systems at finite temperatures. In this manuscript, our focus centers on introducing `MatsubaraFunctions.jl`, a Julia library that implements data structures for generalized $n$-point Green's functions on Matsubara frequency grids. The package's architecture prioritizes user-friendliness without compromising the development of efficient solvers for quantum field theories in equilibrium. Following a comprehensive introduction of the fundamental types, we delve into a thorough examination of key facets of the interface. This encompasses avenues for accessing Green's functions, techniques for extrapolation and interpolation, as well as the incorporation of symmetries and a variety of parallelization strategies. Examples of increasing complexity serve to demonstrate the practical utility of the library, supplemented by discussions on strategies for sidestepping impediments to optimal performance.**

# 1  Motivation

In condensed matter physics, strongly correlated electrons emerge as paradigmatic examples of quantum many-body systems that defy a description in terms of simple band theory, due to their strong interactions with each other and with the atomic lattice. Their study has led to a cascade of discoveries, ranging from high-temperature superconductivity in copper oxides (*cuprates*) [1, 2] to the Mott metal-insulator transition in various condensed matter systems such as, e.g., transition metal oxides or transition metal chalcogenides [3–5] and the emergence of quantum spin liquids in frustrated magnets [6, 7], to name but a few.

The study of correlated electron systems is equally exciting and challenging, not only because the construction of accurate theoretical models requires the consideration of many different degrees of freedom, such as spin, charge, and orbital degrees of freedom, as well as disorder and frustration, but also because of the scarcity of exactly solvable reference Hamiltonians. The single-band Hubbard model in more than one dimension, for example, has remained at the forefront of computational condensed matter physics for decades, although it in many respects can be regarded as the simplest incarnation of a realistic correlated electron system [8, 9]. It is therefore not surprising that a plethora of different numerical methods have been developed to deal with these models [10].

However, no single algorithm is capable of accurately describing all aspects of these complex systems: each algorithm has its strengths and weaknesses, and the choice of algorithm usually depends on the specific problem under investigation. For example, some algorithms, such as *exact diagonalization* (ED) [11–13] or the *density matrix renormalization group* (DMRG) [14, 15] are better suited for studying ground-state properties, while others (*quantum Monte Carlo* (QMC) simulations [16–19], *functional renormalization group* (fRG)

calculations [20–22], ...) perform better when one is interested in dynamic properties such as transport or response functions.

Another popular method, dynamical mean-field theory (DMFT) has been immensely successful; in particular it correctly predicts the Mott transition in the Hubbard model [23]. By approximating the electron self-energy to be local, it however disregards non-local correlation effects, leading to a violation of the Mermin-Wagner theorem [24, 25] as well as a failure to predict the pseudo-gap in the Hubbard model [10]. Non-local (e.g. cluster [26–29] or diagrammatic [30]) extensions of DMFT improve on that front, but are computationally much more expensive. Ultimately, the choice of algorithm is guided by the computational resources available and the trade-off between accuracy and efficiency, as well as by physical insights into which approximations may be justified more than others.

A common motif of many of these algorithms is that they rely on the computation of $n$-particle *Green's functions*, where usually $n = 1, 2$. Roughly speaking, these functions describe correlations within the physical system of interest, such as its response to an external perturbation. In thermal equilibrium, Green's functions are usually defined as imaginary-time-ordered correlation functions, which allows the use of techniques and concepts from statistical mechanics, such as the partition function and free energy. In Fourier space, the corresponding frequencies take on discrete and complex values. This *Matsubara* formalism is widely used to study strongly correlated electron systems, where it provides a powerful tool for calculating thermodynamic quantities, such as the specific heat and magnetic susceptibility, as well as dynamical properties, such as the electron self-energy and optical conductivity [31, 32].

In this manuscript, we present `MatsubaraFunctions.jl`, a software package written in Julia [33] that implements containers for Green's functions in thermal equilibrium. More specifically, it provides a convenient interface for quickly prototyping algorithms involving multivariable Green's functions of the form $G_{i_1...i_n}(\omega_1, ..., \omega_m)$, with lattice/orbital indices $i_k$ ($k = 1, ..., n$) and Matsubara frequencies $\omega_l$ ($l = 1, ..., m$). In an attempt to mitigate monilithic code design and superfluous code reproduction, our goal is to promote a common interface between algorithms where these types of functions make up the basic building blocks. We implement this interface in Julia, since some more recently developed methods, such as the pseudofermion [34–41] and pseudo-Majorana fRG [42–45], seem to have been implemented in Julia as the preferred programming language. In the spirit of similar software efforts, such as the `TRIQS` library for C++ [46], this package therefore aims to provide a common foundation for these and related codes in Julia that is fast enough to facilitate large-scale computations on high-performance computing architectures [47], while remaining flexible and easy to use.

## 2 Equilibrium Green's functions

In this section, we give a brief introduction to equilibrium Green's functions and their properties. In its most general form, an imaginary time, $n$-particle Green's function $G^{(n)}$ is defined as the correlator [48]

$$G^{(n)}_{i_1...i_{2n}}(\tau_1, ..., \tau_{2n}) = \langle \hat{T} c^\dagger_{i_1}(\tau_1) c_{i_2}(\tau_2) ... c^\dagger_{i_{2n-1}}(\tau_{2n-1}) c_{i_{2n}}(\tau_{2n}) \rangle, \tag{1}$$

where $\hat{T}$ is the imaginary-time-ordering operator and $\langle \hat{O} \rangle = \frac{1}{Z} \mathrm{Tr}(e^{-\beta \hat{\mathcal{H}}} \hat{O})$ denotes the thermal expectation value of an operator $\hat{O}$ with respect to the Hamiltonian $\hat{\mathcal{H}}$ at temperature $T = 1/\beta$. Note that natural units are used throughout, in particular we set $k_B \equiv 1$. Here, $c^{(\dagger)}$ are

fermionic or bosonic creation and annihilation operators and $Z = \text{Tr}(e^{-\beta \hat{\mathcal{H}}})$ is the partition function. The indices $i_k$ represent additional degrees of freedom such as lattice site, spin and orbital index. In order for the right-hand side in Eq. (1) to be well defined, it is necessary to restrict the $\tau$ arguments to an interval of length $\beta$, as can be seen, for example, from a spectral (*Lehmann*) representation of the expectation value [48]. Furthermore, the cyclicity of the trace implies that the field variables are anti-periodic in $\beta$ for fermions, or periodic in $\beta$ for bosons, respectively. This allows us to define their Fourier series expansion

$$c_i(\tau) = \frac{1}{\beta} \sum_{\nu_k} c_{i,k}\, e^{-i\nu_k \tau} \qquad\qquad \bar{c}_i(\tau) = \frac{1}{\beta} \sum_{\nu_k} \bar{c}_{i,k}\, e^{i\nu_k \tau} \qquad (2)$$

$$c_{i,k} = \int_0^\beta d\tau\, c_i(\tau)\, e^{i\nu_k \tau} \qquad\qquad \bar{c}_{i,k} = \int_0^\beta d\tau\, \bar{c}_i(\tau)\, e^{-i\nu_k \tau} \qquad (3)$$

where $\nu_k = \frac{\pi}{\beta} \begin{cases} 2k+1 \\ 2k \end{cases}$ , with $k \in \mathbb{Z}$ are the fermionic or bosonic Matsubara frequencies[1].

These definitions carry over to the $n$-particle Green's function $G^{(n)}$, giving

$$G^{(n)}_{i_1...i_{2n}}(\tau_1, ..., \tau_{2n}) = \frac{1}{\beta} \sum_{\nu_1} e^{i\nu_1 \tau_1} ... \frac{1}{\beta} \sum_{\nu_{2n}} e^{-i\nu_{2n} \tau_{2n}} G^{(n)}_{i_1...i_{2n}}(\nu_1, ..., \nu_{2n}) \qquad (4)$$

$$G^{(n)}_{i_1...i_{2n}}(\nu_1, ..., \nu_{2n}) = \int_0^\beta d\tau_1\, e^{-i\nu_1 \tau_1} ... \int_0^\beta d\tau_{2n}\, e^{i\nu_{2n} \tau_{2n}} G^{(n)}_{i_1...i_{2n}}(\tau_1, ..., \tau_{2n})\,. \qquad (5)$$

## 3   Code structure

`MatsubaraFunctions.jl` is an open-source project distributed via Github [49] and licensed under the MIT license. Using Julia's built-in package manager, the code can be easily installed using

```
1 $ julia
2 julia> ]
3 pkg> add https://github.com/dominikkiese/MatsubaraFunctions.jl
```

from the terminal. Here, `]` activates the package manager from the Julia REPL, where `add` downloads the code and its dependencies. The following is an overview of the functionality of the package, starting with a discussion of its basic types and how to use them. A full documentation of the package is available from the github repository.

### 3.1   Basic types

The package evolves around three concrete Julia types: `MatsubaraFrequency`, `MatsubaraGrid` and `MatsubaraFunction`. A Matsubara frequency can be either fermionic or bosonic, that is, $\nu_k = \frac{\pi}{\beta}(2k+1)$ or $\nu_k = \frac{2\pi}{\beta}k$. For a given temperature $T = 1/\beta$ and Matsubara index $k$ they can be constructed using

---

[1]This way, $e^{i\beta\nu_k} = -1$ for fermions and $e^{i\beta\nu_k} = +1$ for bosons such that anti-periodicity or periodicity, respectively, of $c_i(\tau)$ are ensured.

```
1 v = MatsubaraFrequency(T, k, Fermion)
2 w = MatsubaraFrequency(T, k, Boson)
```

Basic arithmetic operations on these objects include addition, subtraction and sign reversal, each of which creates a new `MatsubaraFrequency` instance.

```
1 v1 = v + v # type(v1) = :Boson
2 v2 = w - v # type(v2) = :Fermion
3 v3 = -v    # type(v3) = :Fermion
```

`MatsubaraGrid`s are implemented as sorted collections of uniformly (and symmetrically) spaced Matsubara frequencies. To construct them, users need only specify the temperature, number of positive frequencies, and the particle type.

```
1 T  = 1.0
2 N  = 128
3 g1 = MatsubaraGrid(T, N, Fermion) # total no. frequencies is 2N
4 g2 = MatsubaraGrid(T, N, Boson)   # total no. frequencies is 2N - 1
```

Note that the bosonic Matsubara frequency at zero is included in the positive frequency count. Grid instances are iterable

```
1 for v in g1
2   println(value(v))
3   println(index(v))
4 end
```

and can be evaluated using either a `MatsubaraFrequency` or `Float64` as input.

```
1 idx = rand(eachindex(g1))
2 @assert g1(g1[idx]) == idx
3 @assert g1(value(g1[idx])) == idx
```

Here, we first select a random linear index `idx` and then evaluate `g1` using either the corresponding Matsubara frequency `g1[idx]` or its value. In the former case, `g1(g1[idx])` returns the corresponding linear index of the frequency in the grid, whereas `g1(value(g1[idx]))` finds the linear index of the closest mesh point[2]. The package supports storage of grid instances in H5 file format.

```
1 using HDF5
2 file = h5open("save_g1.h5", "w")
3 save_matsubara_grid!(file, "g1", g1)
4 g1p = load_matsubara_grid(file, "g1")
5 close(file)
```

Finally, a `MatsubaraFunction` is a collection of Matsubara grids with an associated tensor structure $G_{i_1 \dots i_n}$ for each point $(\nu_1, \dots, \nu_m)$ in the Cartesian product of the grids. The indices $i_k$ could, for example, represent lattice sites or orbitals. To construct a `MatsubaraFunction` users need to provide a tuple of `MatsubaraGrid` objects, as well as the dimension of each $i_k$.

---

[2]In both cases the argument must be in bounds, otherwise an exception is thrown.

```
1 T = 1.0
2 N = 128
3 g = MatsubaraGrid(T, N, Fermion)
4
5 # 1D grid, rank 1 tensor with index dimension 1 (scalar valued)
6 f1_complex = MatsubaraFunction(g, 1)
7 f1_real    = MatsubaraFunction(g, 1, Float64)
8
9 # 1D grid, rank 1 tensor with index dimension 5 (vector valued)
10 f2_complex = MatsubaraFunction(g, 5)
11 f2_real    = MatsubaraFunction(g, 5, Float64)
12
13 # 1D grid, rank 2 tensor with index dimension 5 (matrix valued)
14 f3_complex = MatsubaraFunction(g, (5, 5))
15 f3_real    = MatsubaraFunction(g, (5, 5), Float64)
16
17 # 2D grid, rank 2 tensor with index dimension 5 (matrix valued)
18 f4_complex = MatsubaraFunction((g, g), (5, 5))
19 f4_real    = MatsubaraFunction((g, g), (5, 5), Float64)
```

In addition, a floating point type can be passed to the constructor, which fixes the data type for the underlying multidimensional array[3]. Similar to the grids, MatsubaraFunctions can be conveniently stored in H5 format.

```
1 using HDF5
2 file = h5open("save_f1_complex.h5", "w")
3 save_matsubara_function!(file, "f1_complex", f1_complex)
4 f1p = load_matsubara_function(file, "f1_complex")
5 close(file)
```

## 3.2 Accessing and assigning Green's function data

The library provides two possible ways to access the data of a MatsubaraFunction, using either the bracket ([]) or parenthesis (()) operator. While the notion of the former is that of a Base.getindex, the latter evaluates the MatsubaraFunction for the given arguments in such a way that its behavior is well-defined even for out-of-bounds access. The bracket can be used with a set of MatsubaraFrequency instances and tensor indices $i_k$, as well as with Cartesian indices for the underlying data array. It returns the value of the data exactly for the given input arguments, throwing an exception if they are not in bounds. In addition, the bracket can be used to assign values to a MatsubaraFunction as shown in the following example.

```
1 y = 0.5
2 T = 1.0
3 N = 128
4 g = MatsubaraGrid(T, N, Fermion)
5 f = MatsubaraFunction(g, 1)
6
7 for v in g
8     # if there is only one index of dimension 1, it does not need
9     # to be specified, i.e. f[v] can be used instead of f[v, 1]
10    # (also works for the '()' operator)
11    f[v] = 1.0 / (im * value(v) - y)
12 end
```

---

[3]By default, ComplexF64 is used.

```
13
14 # access MatsubaraFunction data
15 v = g[rand(eachindex(g))]
16 println(f[v]) # fast data access, throws error if v is out of bounds
```

When `f` is evaluated using Matsubara frequencies within its grid, it returns the same result as if a bracket was used. However, if the frequencies are replaced by `Float64` values, a multilinear interpolation within the Cartesian product of the grids is performed. If the frequency / float arguments are out of bounds, `MatsubaraFunction`s falls back to extrapolation. The extrapolation algorithm distinguishes between one-dimensional and multidimensional frequency grids. In the 1D case, an algebraic decay is fitted to the high-frequency tail of the `MatsubaraFunction`, which is then evaluated for the given arguments. The functional form of the asymptote is currently restricted to $f(\nu) = \alpha_0 + \frac{\alpha_1}{\nu} + \frac{\alpha_2}{\nu^2}$ (with $\alpha_0, \alpha_1, \alpha_2 \in \mathbb{C}$)[4], which is motivated by the linear or quadratic decay that physical Green's functions typically exhibit. For multidimensional grids, a constant extrapolation is performed from the boundary. Different modes of evaluation are illustrated in an example below.

```
1  y = 0.5
2  T = 1.0
3  N = 128
4  g = MatsubaraGrid(T, N, Fermion)
5  f = MatsubaraFunction(g, 1)
6
7  for v in g
8      f[v] = 1.0 / (im * value(v) - y)
9  end
10
11 # access MatsubaraFunction data
12 v = g[rand(eachindex(g))]
13 println(f(v))        # fast data access, defined even if v is out of bounds
14 println(f(value(v))) # slow data access, uses interpolation
15
16 # polynomial extrapolation in 1D, constant term set to 0 (the default)
17 vp = MatsubaraFrequency(T, 256, Fermion)
18 println(f(vp; extrp = ComplexF64(0.0)))
```

### 3.3 Extrapolation of Matsubara sums

A common task when working with equilibrium Green's functions is the calculation of Matsubara sums $\frac{1}{\beta} \sum_\nu f(\nu)$, where we have omitted additional indices of $f$ for brevity. However, typical Green's functions decay rather slowly (algebraically) for large frequencies, which presents a technical difficulty for the accurate numerical calculation of their Matsubara sums: they may require some regulator function to control the convergence[5] (difficult to implement) and a large number of frequencies to sum over (expensive). In contrast, there exist analytical results for simple functional forms of $f$ even in cases where a straightforward numerical summation fails. `MatsubaraFunctions` provides the `sum_me` function, which can be used to calculate sums over complex-valued $f(\nu)$, if $f(z)$ (with $z \in \mathbb{C}$) decays to zero for large $|z|$ and is representable by a Laurent series in an elongated annulus about the imaginary axis (see App. A for details). An example for its use is shown below. Note that this feature is experimental and its API as well as the underlying algorithm might change in future versions.

---

[4]Note that $\alpha_0$ has to be provided by the user.

[5]For example, a factor $e^{i\nu 0^\pm}$ might be necessary in cases where $f$ decays linearly in $\nu$.

```
1  y = 0.5
2  T = 1.0
3  N = 128
4  g = MatsubaraGrid(T, N, Fermion)
5  f = MatsubaraFunction(g, 1)
6
7  for v in g
8      f[v] = 1.0 / (im * value(v) - y)
9  end
10
11 # evaluate the series and compare to analytic result
12 rho(x, T) = 1.0 / (exp(x / T) + 1.0)
13 println(abs(sum_me(f) - (rho(+y, T) - 1.0)))
```

## 3.4  Padé approximants

Although the Matsubara formalism provides a powerful tool for the calculation of thermodynamic quantities, it lacks the ability to directly determine, for example, dynamic response functions or transport properties associated with real-frequency Green's functions, which facilitate comparison with experiments. There have been recent advances in the use of real-frequency quantum field theory [50–53], yet the calculation of dynamic real-frequency Green's functions remains a technically challenging endeavor. In many applications, therefore, one resorts to calculations on the imaginary axis and then performs an analytic continuation in the complex upper half-plane to determine observables on the real-frequency axis. The analytic continuation problem is ill-conditioned, because there may be significantly different real-frequency functions describing the same set of complex-frequency data within finite precision. Nevertheless, there has been remarkable progress in the development of numerical techniques such as the maximum entropy method [54–56] or stochastic analytical continuation [57, 58]. These methods are particularly useful when dealing with stochastic noise induced by Monte Carlo random sampling. A corresponding implementation in Julia is, for example, provided by the `ACFlow` toolkit [59]. On the other hand, if the input data are known with a high degree of accuracy (as in the fRG and related approaches), analytic continuation using Padé approximants is a valid alternative. Here, one first fits a rational function to the complex frequency data which is then used as a proxy for the Green's function in the upper half-plane. If the function of interest has simple poles this procedure can already provide fairly accurate results, see e.g. Ref. [60]. In `MatsubaraFunctions` we implement the fast algorithm described in the appendix of Ref. [61], which computes an $N$-point Padé approximant for a given set of data points $\{(x_i, y_i)|i = 1, ..., N\}$. A simple example of its use is shown below. Note that it might be necessary to use higher precision floating-point arithmetic to cope with rounding errors in the continued fraction representation used for calculating the Padé approximant.

```
1  # some dummy function
2  as   = ntuple(x -> rand(BigFloat), 4)
3  f(x) = as[1] / (1.0 + as[2] * x / (1.0  + as[3] * x / (1.0 + as[4] * x)))
4
5  # generate sample and compute Pade approximant
6  xdata = Vector{BigFloat}(0.01 : 0.01 : 1.0)
7  ydata = f.(xdata)
8  PA    = PadeApprox(xdata, ydata)
9
10 @assert length(coeffs(PA)) == 5
11 @assert PA.(xdata) ≈ ydata
```

### 3.5 Automated symmetry reduction

In many cases, the numerical effort of computing functions in the Matsubara domain can be drastically reduced by the use of symmetries. For one-particle fermionic Green's functions $G_{i_1 i_2}(\nu)$, for example, complex conjugation implies that $G_{i_1 i_2}(\nu) = G^*_{i_2 i_1}(-\nu)$, relating positive and negative Matsubara frequencies. Our package provides an automated way to compute the set of irreducible `MatsubaraFunction` components[6], given a list of one or more symmetries as is illustrated in the following example

```
1 y = 0.5
2 T = 1.0
3 N = 128
4 g = MatsubaraGrid(T, N, Fermion)
5 f = MatsubaraFunction(g, 1)
6
7 for v in g
8     f[v] = 1.0 / (im * value(v) - y)
9 end
10
11 # complex conjugation acting on Green's function
12 function conj(
13     w :: Tuple{MatsubaraFrequency},
14     x :: Tuple{Int64}
15     ) :: Tuple{Tuple{MatsubaraFrequency}, Tuple{Int64}, MatsubaraOperation}
16
17     return (-w[1],), (x[1],), MatsubaraOperation(sgn = false, con = true)
18 end
19
20 # compute the symmetry group
21 SG = MatsubaraSymmetryGroup([MatsubaraSymmetry{1, 1}(conj)], f)
22
23 # obtain another Green's function by symmetrization
24 function init(
25     w :: Tuple{MatsubaraFrequency},
26     x :: Tuple{Int64}
27     ) :: ComplexF64
28
29     return f[w, x...]
30 end
31
32 InitFunc = MatsubaraInitFunction{1, 1, ComplexF64}(init)
33 h        = MatsubaraFunction(g, 1)
34 SG(h, InitFunc)
35 @assert h == f
```

Here, one first constructs an instance of type `MatsubaraSymmetry` by passing a function that maps the input arguments of `f` to new arguments extended by a `MatsubaraOperation`. The latter specifies whether the evaluation of `f` on the mapped arguments should be provided with an additional sign or complex conjugation. Next, the irreducible array elements are computed and an object of type `MatsubaraSymmetryGroup`[7] is constructed from a vector of symmetries provided by the user. Here, the length of the vector is one (we only considered complex conjugation), but the generalization to multiple symmetries is straightforward (see Ref. [62] for more examples). A `MatsubaraSymmetryGroup` can be called with a `MatsubaraFunction`

---

[6]These are all elements of the underlying data array which cannot be mapped to each other by symmetries.

[7]A `MatsubaraSymmetryGroup` contains all groups of symmetry equivalent elements and the operations needed to map them to each other.

and an initialization function[8]. This call will evaluate the `MatsubaraInitFunction` for all irreducible elements of the symmetry group of `f`, writing the result into the data array of `h`. Finally, all symmetry equivalent elements are determined without additional calls to the (costly) initialization function. Symmetry groups can be stored in H5 format as shown below.

```
1 using HDF5
2 file = h5open("save_SG.h5", "w")
3 save_matsubara_symmetry_group!(file, "SG", SG)
4 SGp = load_matsubara_symmetry_group(file, "SG")
5 close(file)
```

### 3.6  Running in parallel

To simplify code parallelization when using `MatsubaraFunctions.jl`, the package has some preliminary MPI support based on the `MPI.jl` wrapper and illustrated in an example below.

```
1 using MatsubaraFunctions
2 using MPI
3
4 MPI.Init()
5 mpi_info()
6 mpi_println("I print on main.")
7 ismain = mpi_ismain() # ismain = true if rank is 0
8
9 y = 0.5
10 T = 1.0
11 N = 128
12 g = MatsubaraGrid(T, N, Fermion)
13 f = MatsubaraFunction(g, 1)
14
15 for v in g
16     f[v] = 1.0 / (im * value(v) - y)
17 end
18
19 # simple loop parallelization for UnitRange
20 for vidx in mpi_split(1 : length(g))
21   println("My rank is $(mpi_rank()): $(vidx)")
22 end
23
24 # simple (+) allreduce
25 mpi_allreduce!(f)
```

Calls of `MatsubaraSymmetryGroup` with an initialization function have an opt-in switch (`mode`) to enable parallel evaluation of the `MatsubaraInitFunction` (by default `mode = :serial`). If `mode = :polyester`, shared memory multithreading as provided by the `Polyester` Julia package [63] is used[9]. This mode is recommended for initialization functions that are cheap to evaluate and are unlikely to benefit from `Threads.@threads` due to the overhead from invoking the Julia scheduler. For more expensive functions, users can choose between `mode = :threads`, which simply uses `Threads.@threads`, and `mode = :hybrid`. The latter combines both `MPI` and native Julia threads and can therefore be used to run calculations on multiple compute nodes.

---

[8]A `MatsubaraInitFunction` takes a tuple of Matsubara frequencies and tensor indices and returns a floating point type.

[9]Here, the `batchsize` argument can be used to control the number of threads involved.

### 3.7  Performance note

By default, types in `MatsubaraFunctions.jl` perform intrinsic consistency checks when they are invoked. For example, when computing the linear index of a `MatsubaraFrequency` in a `MatsubaraGrid`, we make sure that the particle types and temperatures match between the two. While this ensures a robust *modus operandi*, it unfortunately impacts performance, especially for larger projects. To deal with this issue, we have implemented a simple switch, `MatsubaraFunctions.sanity_checks()`, which, when turned off[10] disables `@assert` expressions. It is not recommended to touch this switch until an application has been thoroughly tested, as it leads to wrong results on improper use. For the MBE solver discussed in Sec. 4.3.2, we found runtime improvements of up to 10% when the consistency checks were disabled.

## 4  Examples

### 4.1  Hartree-Fock calculation in the atomic limit

As a first example of the use of `MatsubaraFunctions.jl` we consider the calculation of the one-particle Green's function $G$ using the Hartree-Fock (HF) approximation in the atomic limit of the Hubbard model, i.e., we consider the Hamiltonian

$$\hat{\mathcal{H}} = U\hat{n}_\uparrow \hat{n}_\downarrow - \mu(\hat{n}_\uparrow + \hat{n}_\downarrow)\,, \tag{6}$$

where $U$ denotes the Hubbard interaction and $\hat{n}_\sigma$ are the density operators for spin up and down. In the following, we fix the chemical potential to $\mu = 0$, i.e., we consider the system in the strongly hole-doped regime.

The Hartree-Fock theory [64–66] is a widespread method in condensed matter physics used to describe, e.g., electronic structures and properties of materials [67, 68]. It is a mean-field approximation as it treats the electrons in a solid as independent particles being subject to an effective background field due to all the other particles.

In an interacting many-body system, the bare Green's function $G_0$ has to be dressed by self-energy insertions, here denoted by $\Sigma$, in order to obtain $G$, which is summarized in the Dyson equation

$$\boldsymbol{G} = \boldsymbol{G}_0[\mathbb{1} - \boldsymbol{\Sigma}\,\boldsymbol{G}_0]^{-1} = \boldsymbol{G}_0 + \boldsymbol{G}_0\boldsymbol{\Sigma}\boldsymbol{G}_0 + \boldsymbol{G}_0\boldsymbol{\Sigma}\boldsymbol{G}_0\boldsymbol{\Sigma}\boldsymbol{G}_0 + ...\,, \tag{7}$$

where $\boldsymbol{G}_0$, $\boldsymbol{G}$ and $\boldsymbol{\Sigma}$ in general are matrix-valued. In HF theory one only considers the lowest order term contributing to the self-energy, which is linear in the interaction potential. For the spin-rotation invariant single-site system at hand, $\boldsymbol{\Sigma} = \Sigma_\sigma = \Sigma$ and the HF approximation for the self-energy reads

$$\Sigma(\nu) \approx \frac{U}{\beta} \sum_{\nu'} G(\nu')e^{i\nu'0^+} = Un\,, \tag{8}$$

where $n$ is the density per spin. The Dyson equation then takes the simple form

$$G(\nu) \approx [G_0^{-1}(\nu) - Un]^{-1}\,. \tag{9}$$

---

[10]`MatsubaraFunctions.sanity_checks() = false`

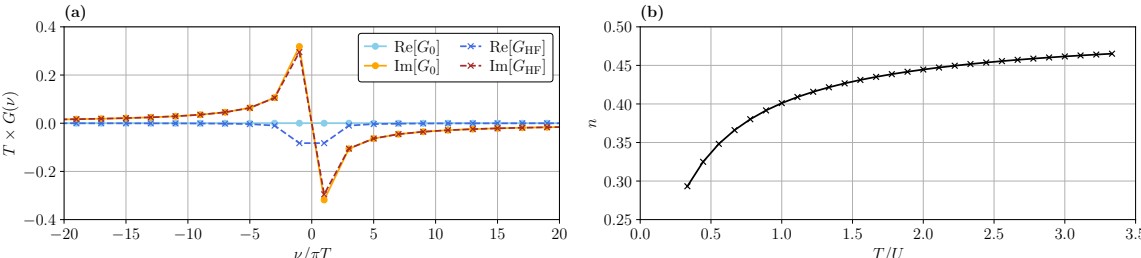

**Figure 1: Exemplary Hartree-Fock results.** (a) Comparison of the bare Green's function $G_0$ with the HF result $G_{\text{HF}}$ for $T/U = \frac{1}{3}$. (b) Hartree-Fock density $n$ as a function of temperature.

Below, we demonstrate how to set up and solve Eqs. (8) & (9) self-consistently for the density $n$ using Anderson acceleration [69, 70] as provided by the `NLsolve` Julia package[71] in conjunction with `MatsubaraFunctions.jl`.

```julia
using MatsubaraFunctions
using NLsolve

const T = 0.3  # temperature
const U = 0.9  # interaction
const N = 1000 # no. frequencies

# initialize Green's function container
g = MatsubaraGrid(T, N, Fermion)
G = MatsubaraFunction(g, 1)

for v in g
    G[v] = 1.0 / (im * value(v))
end

# set up fixed-point equation for NLsolve
function fixed_point!(F, n, G)

    # calculate G
    for v in grids(G, 1)
        G[v] = 1.0 / (im * value(v) - U * n[1])
    end

    # calculate the residue
    F[1] = density(G) - n[1]

    return nothing
end

res = nlsolve((F, n) -> fixed_point!(F, n, G), [density(G)], method = :anderson)
```

Here, we first build the `MatsubaraFunction` container for $G$ and initialize it to $G_0(\nu) = \frac{1}{i\nu}$. This container is then passed to the fixed-point equation using an anonymous function, which mutates $G$ on each call to incorporate the latest estimate of $n^{11}$. Fig. 1 shows exemplary results for the full Green's function and HF density. As can be seen from Fig. 1(b) the latter deviates from its bare value $n_0 = \frac{1}{2}$ when the temperature is decreased and approaches $n = 0$ for $T \to 0$, as expected.

---

[11]Here, we make use of the `density` function, which calculates the Fourier transform $f(\tau \to 0^-)$ given a complex-valued input function $f(\nu)$.

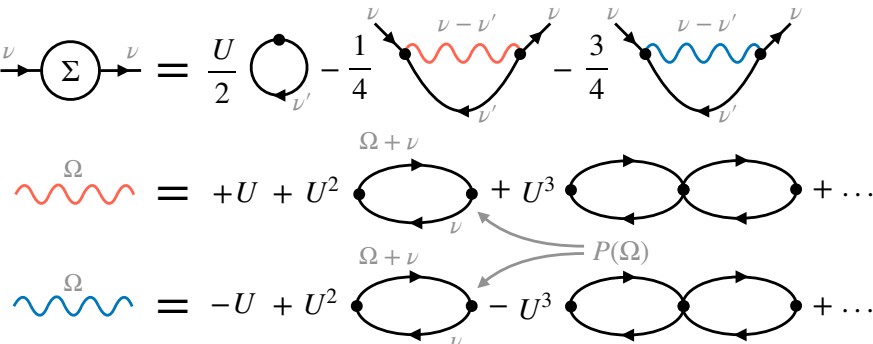

**Figure 2: Diagrammatic representation of spin-conserving $GW$ equations in the atomic limit.** Wavy lines denote the screened interactions in the density (red) and magnetic (blue) channel. They are obtained by dressing the respective bare interactions with a series of bubble diagrams $P(\Omega)$, as illustrated in the second and third line from the top.

## 4.2   $GW$ calculation in the atomic limit

In this section, we extend our Hartree-Fock code to include *bubble* corrections[12] in the calculation of the self-energy. The resulting equations, commonly known as the $GW$ approximation, allow us to exemplify the use of more advanced library features, such as extrapolation of the single-particle Green's function and the implementation of symmetries. Therefore, they present a good starting point for the more involved application discussed in Sec. 4.3.1.

The $GW$ approximation is a widely used method in condensed matter physics and quantum chemistry for calculating electronic properties of materials [72–74]. In addition to the Hartree term $\Sigma_{\mathrm{H}} = Un$, which considers only the bare interaction, the mutual screening of the Coulomb interaction between electrons is partially taken into account. For spin-rotation invariant systems it is common practice to decouple these *screened interactions* $\eta$[13] into a density (or charge) component $\eta^D$ and a magnetic (or spin) component $\eta^M$ (see App. B), such that

$$\Sigma(\nu) \approx \tfrac{Un}{2} - \tfrac{1}{\beta} \sum_{\nu'} G(\nu') \left[ \tfrac{1}{4}\eta^D(\nu - \nu') + \tfrac{3}{4}\eta^M(\nu - \nu') \right] , \tag{10}$$

for the atomic limit Hamiltonian. The $\eta$ are computed by summing a series of bubble diagrams in the particle-hole channel, i.e.,

$$\eta^{D/M}(\Omega) = \frac{\pm U}{1 \mp UP(\Omega)} , \tag{11}$$

where the *polarization bubble* $P$ is given by

$$P(\Omega) = \tfrac{1}{\beta} \sum_{\nu} G(\Omega + \nu)G(\nu) . \tag{12}$$

A diagrammatic representation of these relations is shown in Fig. 2. Finally, the set of equations is closed by computing $G$ from the Dyson equation. Since the Green's function

---

[12]That is, Feynman diagrams formed by a closed loop of two single-particle Green's functions.

[13]Here, we denote the screened interactions by $\eta$ instead of $W$ to avoid conflicting notation with the code examples in Sec. 4.3.2.

transforms as $G^*(\nu) = G(-\nu)$ under complex conjugation [48], we also have that

$$P(-\Omega) = \tfrac{1}{\beta} \sum_\nu G(-\Omega + \nu) G(\nu) = \tfrac{1}{\beta} \sum_\nu G(-\Omega - \nu) G(-\nu) = \tfrac{1}{\beta} \sum_\nu G^*(\Omega + \nu) G^*(\nu)$$

$$= P^*(\Omega)\,, \tag{13}$$

and likewise $\Sigma^*(\nu) = \Sigma(-\nu)$. Thus, the numerical effort for evaluating Eqs. (10) and (12) can be reduced by a factor of two using a `MatsubaraSymmetryGroup` object. To structure the $GW$ code, we first write a solver class which takes care of the proper initialization of the necessary `MatsubaraFunction` instances.

```julia
1  using MatsubaraFunctions
2  using HDF5
3
4  conj(w, x) = (-w[1],), (x[1],), MatsubaraOperation(sgn = false, con = true)
5
6  struct GWsolver
7      T     :: Float64
8      U     :: Float64
9      N     :: Int64
10     G     :: MatsubaraFunction{1, 1, 2, ComplexF64}
11     Sigma :: MatsubaraFunction{1, 1, 2, ComplexF64}
12     P     :: MatsubaraFunction{1, 1, 2, ComplexF64}
13     η_D   :: MatsubaraFunction{1, 1, 2, ComplexF64}
14     η_M   :: MatsubaraFunction{1, 1, 2, ComplexF64}
15     SGf   :: MatsubaraSymmetryGroup
16     SGb   :: MatsubaraSymmetryGroup
17
18     function GWsolver(T, U, N)
19
20         # fermionic containers
21         gf    = MatsubaraGrid(T, N, Fermion)
22         G     = MatsubaraFunction(gf, 1)
23         Sigma = MatsubaraFunction(gf, 1)
24
25         # bosonic containers
26         gb  = MatsubaraGrid(T, N, Boson)
27         P   = MatsubaraFunction(gb, 1)
28         η_D = MatsubaraFunction(gb, 1)
29         η_M = MatsubaraFunction(gb, 1)
30
31         # symmetry groups
32         SGf = MatsubaraSymmetryGroup([MatsubaraSymmetry{1, 1}(conj)], G)
33         SGb = MatsubaraSymmetryGroup([MatsubaraSymmetry{1, 1}(conj)], P)
34
35         return new(T, U, N, G, Sigma, P, η_D, η_M, SGf, SGb)
36     end
37  end
```

As a second step, we implement the self-consistent equations, which we solve using Anderson acceleration. Note that we have rewritten the $GW$ equation for the self-energy as

$$\Sigma(\nu) \approx Un - \tfrac{1}{\beta} \sum_{\nu'} G(\nu') \left[ \tfrac{1}{4} \eta^D(\nu - \nu') + \tfrac{3}{4} \eta^M(\nu - \nu') + \tfrac{U}{2} \right]\,, \tag{14}$$

which is beneficial since the product of $G$ with the constant contributions to $\eta^{D/M}$ simply shifts the real part of the self-energy by $\frac{Un}{2}$ such that $\Sigma = \Sigma_{\mathrm{H}} + \mathcal{O}(U^2)$.

```
 1 function fixed_point!(F, x, S)
 2
 3     # update Sigma
 4     unflatten!(S.Sigma, x)
 5
 6     # calculate G
 7     for v in grids(S.G, 1)
 8         S.G[v] = 1.0 / (im * value(v) - S.Sigma[v])
 9     end
10
11     sum_grid = MatsubaraGrid(S.T, 4 * S.N, Fermion)
12
13     # calculate P using symmetries
14     function calc_P(wtpl, xtpl)
15
16         P = 0.0
17
18         for v in sum_grid
19             P += S.G(v + wtpl[1]) * S.G(v)
20         end
21
22         return S.T * P
23     end
24
25     S.SGb(S.P, MatsubaraInitFunction{1, 1, ComplexF64}(calc_P))
26
27     # calculate η_D and η_M
28     for w in S.P.grids[1]
29         S.η_D[w] = +S.U / (1.0 - S.U * S.P[w])
30         S.η_M[w] = -S.U / (1.0 + S.U * S.P[w])
31     end
32
33     # calculate Sigma using symmetries
34     function calc_Sigma(wtpl, xtpl)
35
36         Sigma = S.U * density(S.G)
37
38         for v in sum_grid
39             Sigma -= S.T * S.G(v) * (
40                 0.25 * S.η_D(wtpl[1] - v; extrp = ComplexF64(+S.U)) +
41                 0.75 * S.η_M(wtpl[1] - v; extrp = ComplexF64(-S.U)) +
42                 0.50 * S.U)
43         end
44
45         return Sigma
46     end
47
48     S.SGf(S.Sigma, MatsubaraInitFunction{1, 1, ComplexF64}(calc_Sigma))
49
50     # calculate the residue
51     flatten!(S.Sigma, F)
52     F .-= x
53
54     return nothing
55 end
```

Here, we make use of the `flatten!` and `unflatten!` functions which allow us to parse `MatsubaraFunction` data into a one dimensional array[14]. The fixed-point can now easily be computed with, for example,

```
 1 const T = 0.3  # temperature
```

---

[14]We also export `flatten` which will allocate a new 1D array from the `MatsubaraFunction`.

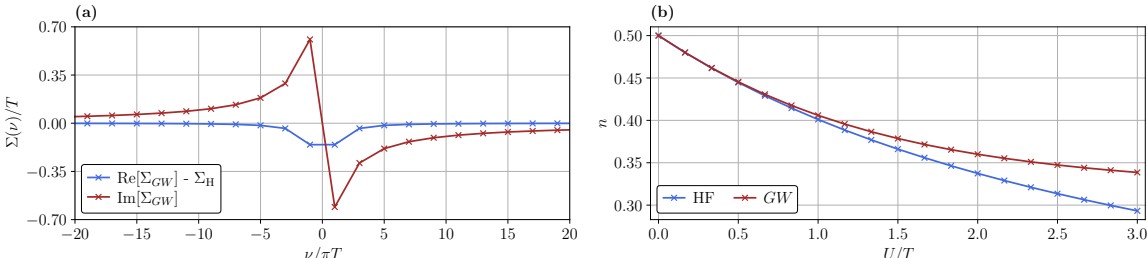

**Figure 3: Exemplary $GW$ results.** (a) The complex-valued self-energy $\Sigma_{GW}$ with its real part offset by the Hartree shift $\Sigma_{\mathrm{H}} = U n_{GW}$ for $T/U = \frac{1}{3}$. (b) $GW$ and Hartree-Fock densities as a function of $U/T$.

```
2 const U = 0.9  # interaction
3 const N = 1000 # no. frequencies
4
5 S    = GWsolver(T, U, N)
6 init = zeros(ComplexF64, length(S.Sigma))
7 res  = nlsolve((F, x) -> fixed_point!(F, x, S), init, method = :anderson, m = 8, beta = 0.5, show_trace
  ↪    = true)
```

In Fig. 3 we show exemplary results for the self-energy and density obtained in $GW$. In contrast to the Hartree-Fock calculations in the previous section, $\Sigma$ is now a frequency-dependent quantity, whose real part asymptotically approaches $U n_{GW}$. As can be seen from Fig. 3(b), these $GW$ densities agree quantitatively with the HF result for weak interactions $U/T \lesssim \frac{1}{2}$, but yield larger densities for higher values of $U$ as expected when the local interaction is partially screened.

## 4.3 Multiboson exchange solver for the single impurity Anderson model

*Note: Readers who are not interested in the formal discussion presented below should feel free to skip this section and proceed directly to Section 5 on future directions.*

In the following, we extend upon the previous computations for the Hubbard atom by coupling the single electronic level to a bath of non-interacting electrons. Specifically, we consider the *single-impurity Anderson model*, a minimal model for localized magnetic impurities in metals introduced by P.W. Anderson to explain the physics behind the Kondo effect [75]. It is defined by the Hamiltonian

$$H = \sum_{\sigma} \epsilon_{\mathrm{d}} d_{\sigma}^{\dagger} d_{\sigma} + U n_{\mathrm{d},\uparrow} n_{\mathrm{d},\downarrow} + \sum_{k,\sigma} \left( V_k d_{\sigma}^{\dagger} c_{k,\sigma} + V_k^* c_{k,\sigma}^{\dagger} d_{\sigma} \right) + \sum_{k,\sigma} \epsilon_{k,\sigma} c_{k,\sigma}^{\dagger} c_{k,\sigma} \,, \qquad (15)$$

describing an impurity $d$ level $\epsilon_{\mathrm{d}}$, hybridized with conduction electrons of the metal via a matrix element $V_k$. The electrons in the localized $d$ state, where $n_{\mathrm{d},\sigma} = d_{\sigma}^{\dagger} d_{\sigma}$, interact according to the interaction strength $U$, whereas the $c$ electrons of the bath are non-interacting. Following [76], in a path-integral formulation for the partition function $Z = \int \prod_{\sigma} \mathcal{D}\left(\bar{d}_{\sigma}\right) \mathcal{D}\left(d_{\sigma}\right) \mathcal{D}\left(\bar{c}_{k,\sigma}\right) \mathcal{D}\left(c_{k,\sigma}\right) \mathrm{e}^{-S}$ with the action $S = \int_0^{\beta} \mathcal{L}(\tau) \mathrm{d}\tau$, the Lagrangian for

the model is given by

$$\mathcal{L}(\tau) = \sum_\sigma \bar{d}_\sigma(\tau)\left(\partial_\tau + \epsilon_\mathrm{d}\right) d_\sigma(\tau) + \sum_{k,\sigma} \bar{c}_{k,\sigma}(\tau)\left(\partial_\tau + \epsilon_k\right) c_{k,\sigma}(\tau)$$
$$+ U n_\uparrow(\tau) n_\downarrow(\tau) + \sum_{k,\sigma} V_k \left[\bar{d}_\sigma(\tau) c_{k,\sigma}(\tau) + \bar{c}_{k,\sigma}(\tau) d_\sigma(\tau)\right] , \tag{16}$$

where $n_\sigma(\tau) = \bar{d}_\sigma(\tau) d_\sigma(\tau)$. Integrating over the only quadratically occurring Grassmann variables for the bath electrons, one formally obtains $Z = \int \prod_\sigma \mathcal{D}\left(\bar{d}_\sigma\right) \mathcal{D}\left(d_\sigma\right) \mathrm{e}^{-S_\mathrm{red}}$ with the reduced action given by

$$S_\mathrm{red} = \int_0^\beta \mathrm{d}\tau \int_0^\beta \mathrm{d}\tau' \sum_\sigma \bar{d}_\sigma(\tau) \left[-G_\sigma^{(0)}\left(\tau - \tau'\right)\right]^{-1} d_\sigma\left(\tau'\right) + U \int_0^\beta \mathrm{d}\tau\, n_\uparrow(\tau) n_\downarrow(\tau) . \tag{17}$$

Switching to Matsubara frequencies as described in section 2, the non-interacting Green's function for the localized $d$ electrons reads

$$G_\sigma^{(0)}\left(\nu_n\right) = \frac{1}{\mathrm{i}\nu_n - \epsilon_\mathrm{d} + \Delta(\nu_n)} . \tag{18}$$

Following [77] we choose an isotropic hybridization strength $V_k \equiv V$ and a flat density of states with bandwidth $2D$ for the bath electrons, leading to the hybridization function[15] $\Delta(\nu_n) = \mathrm{i}\frac{V^2}{D} \arctan\frac{D}{\nu_n}$. In the following, we set $V = 2$, measure energy in units of $V/2 = 1$ and set the half bandwidth to $D = 10$. In the context of this work, we focus on the particle-hole symmetric model, setting $\epsilon_\mathrm{d} = -U/2$. Then, the Hartree term of the self-energy, $\Sigma_\mathrm{H} = U/2$ is conveniently absorbed into the bare propagator,

$$G_\sigma^{(0)}(\nu_n) \to G_\sigma^\mathrm{H}(\nu_n) = \frac{1}{\mathrm{i}\nu_n - \epsilon_\mathrm{d} + \Delta(\nu_n) - \Sigma_\mathrm{H}} = \frac{1}{\mathrm{i}\nu_n + \Delta(\nu_n)} . \tag{19}$$

Consequently, the Hartree propagator is used instead of the bare propagator throughout.

### 4.3.1 Single boson exchange decomposition of the parquet equations

Following [78], we now reiterate the single-boson exchange (SBE) decomposition of the four-point vertex and, subsequently, of the parquet equations. The starting point for the SBE decomposition, which was originally developed in [79–84], is the unambiguous classification of vertex diagrams according to their $U$-reducibility in each channel. In order to introduce this concept in the context of the parquet equations, we first have to discuss the similar concept of two-particle reducibility, which provides the basis for the parquet decomposition of the vertex,

$$\Gamma = \Lambda_\mathrm{2PI} + \gamma_a + \gamma_p + \gamma_t. \tag{20}$$

This decomposition states that all diagrams which contribute to the two-particle vertex $\Gamma$ can be classified as being part of one of four disjoint contributions: $\gamma_r$ with $r \in \{a, p, t\}$ collects those diagrams which are two-particle reducible (2PR) in channel $r$, i.e., they can be disconnected by cutting a pair of propagator lines, which can either be aligned in an

---

[15]Note that we use a different sign convention for the hybridization function compared to [77].

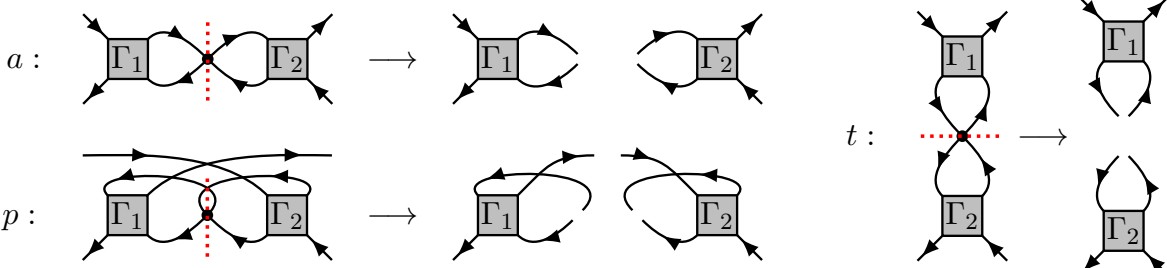

**Figure 4: Illustration of $U$-reducibility in the three two-particle channels $a$, $p$ and $t$.** The Figure is analogous to Fig. 4 of [80] and adapted from [78]. $\Gamma_1$ and $\Gamma_2$ can be any vertex diagram or the unit vertex.

antiparallel ($a$), parallel ($p$) or transverse antiparallel ($t$) way. All remaining diagrams, which are not 2PR in either of the three channels, contribute to $\Lambda_{2\text{PI}}$, the fully two-particle irreducible (2PI) vertex. One can equally well decompose $\Gamma$ w.r.t. its two-particle reducibility in one of the three channels, $\Gamma = I_r + \gamma_r$, which defines $I_r = \Lambda_{2\text{PI}} + \sum_{r' \neq r} \gamma_{r'}$, collecting all diagrams that are 2PI in channel $r$. The Bethe-Salpeter equations (BSEs) then relate the reducible diagrams to the irreducible ones,

$$\gamma_r = I_r \circ \Pi_r \circ \Gamma = \Gamma \circ \Pi_r \circ I_r. \tag{21}$$

This short-hand notation introduces the $\Pi_r$ bubble, i.e., the propagator pair connecting the two vertices, see [78] for their precise channel-dependent definition, as well as for the connector symbol $\circ$, which channel-dependently denotes summation over internal frequencies and quantum numbers. The self-energy $\Sigma$, which enters the propagator via the Dyson equation $G = G_0 + G_0 \Sigma G$, is provided by the Schwinger-Dyson equation (SDE),

$$\Sigma = -\left(U + U \circ \Pi_p \circ \Gamma\right) \cdot G = -\left(U + \tfrac{1}{2} U \circ \Pi_a \circ \Gamma\right) \cdot G\,, \tag{22}$$

where $U$ is the bare interaction and the symbol $\cdot$ denotes the contraction of two vertex legs with a propagator. Together, equations (20), (21) and (22) are known as the *parquet equations* [85, 86] and can be solved self-consistently, if the 2PI vertex $\Lambda_{2\text{PI}}$ is provided [87–90]. Unfortunately, $\Lambda_{2\text{PI}}$ is the most complicated object, as its contributions contain nested contractions over internal arguments. Often, the parquet approximation (PA) is therefore employed, which truncates $\Lambda_{2\text{PI}}$ beyond the bare interaction $U$. In the context of the SBE decomposition relevant to this work, $U$-reducibility is an alternative criterion to the concept of two-particle reducibility for the classification of vertex diagrams. A diagram that is 2PR in channel $r$ is also said to be $U$-reducible in channel $r$ if it can be disconnected by removing one bare vertex that is attached to a $\Pi_r$ bubble, as illustrated in Fig. 4. Furthermore, the bare vertex $U$ is defined to be $U$-reducible in all three channels. The $U$-reducible diagrams in channel $r$ are in the following denoted $\nabla_r$ and are said to describe *single-boson exchange* processes, as the linking bare interaction $U$, which would disconnect the diagram if cut, mediates just a single bosonic transfer frequency. The diagrams which are 2PR in channel $r$ but not $U$-reducible in channel $r$ are called *multi-boson exchange* diagrams and denoted $M_r$. With these classifications, the two-particle reducible vertices can be written as $\gamma_r = \nabla_r - U + M_r$, making sure to exclude $U$, which is contained in $\nabla_r$ but not in $\gamma_r$. The parquet decomposition (20) yields in this language,

$$\Gamma = \varphi^{U\text{irr}} + \sum_r \nabla_r - 2U, \qquad\qquad \varphi^{U\text{irr}} = \Lambda_{2\text{PI}} - U + \sum_r M_r\,, \tag{23}$$

where $\varphi^{U\mathrm{irr}}$ is the fully $U$-irreducible part of $\Gamma$. For a diagrammatic illustration of the first equation, see Fig. 8 in [78]. The channel-dependent decomposition of the vertex $\Gamma = I_r + \gamma_r = \nabla_r + T_r$ can also be split into $U$-reducible and $U$-irreducible parts in channel $r$, defining the $U$-irreducible remainder $T_r = I_r - U + M_r$ in channel $r$. Inserting all these definitions into the BSEs (21) and separating $U$-reducible and $U$-irreducible contributions gives the two sets of equations,

$$\nabla_r - U = I_r \circ \Pi_r \circ \nabla_r + U \circ \Pi_r \circ T_r = \nabla_r \circ \Pi_r \circ I_r + T_r \circ \Pi_r \circ U, \tag{24}$$

$$M_r = (I_r - U) \circ \Pi_r \circ T_r = T_r \circ \Pi_r \circ (I_r - U), \tag{25}$$

for each channel $r$. From equation (24) one can derive (see [78] for details) that the single-boson exchange terms can be written as $\nabla_r = \bar{\lambda}_r \bullet \eta_r \bullet \lambda_r$, where $\bar{\lambda}_r, \lambda_r$ denote the *Hedin vertices* [72] and $\eta_r$ the screened interaction in channel $r$. The former are related to the $U$-irreducible vertex in channel $r$ via $\bar{\lambda}_r = \mathbf{1}_r + T_r \circ \Pi_r \circ \mathbf{1}_r = \mathbf{1}_r + \mathbf{1}_r \circ \Pi_r \circ T_r$ and can be understood as $U$-irreducible, amputated parts of three-point functions, as they depend on only two frequencies. In contrast to the $GW$ approximation discussed in Sec. 4.2, the screened interaction $\eta_r$ is now defined in terms of a Dyson equation, $\eta_r = U + U \bullet P_r \bullet \eta_r = U + \eta_r \bullet P_r \bullet U$, with the polarization $P_r = \lambda_r \circ \Pi_r \circ \mathbf{1}_r = \mathbf{1}_r \circ \Pi_r \circ \bar{\lambda}_r$ dressed by vertex corrections. In the previous expressions, the connector $\bullet$ denotes an internal summation similar to $\circ$, the only difference being that summation over frequencies is excluded. The corresponding unit vertex is denoted $\mathbf{1}_r$.

Lastly, one can rewrite the SDE in terms of the screened interaction and the Hedin vertex in channel $r$ which yields, for example, $-\Sigma = (\eta_p \bullet \lambda_p) \cdot G = (\bar{\lambda}_p \bullet \eta_p) \cdot G$ if one chooses $r = p$.

In summary, the SBE-equations to be solved read

$$\eta_r = U + U \bullet P_r \bullet \eta_r = U + \eta_r \bullet P_r \bullet U, \tag{26a}$$

$$P_r = \lambda_r \circ \Pi_r \circ \mathbf{1}_r = \mathbf{1}_r \circ \Pi_r \circ \bar{\lambda}_r, \tag{26b}$$

$$\bar{\lambda}_r = \mathbf{1}_r + T_r \circ \Pi_r \circ \mathbf{1}_r, \tag{26c}$$

$$\lambda_r = \mathbf{1}_r + \mathbf{1}_r \circ \Pi_r \circ T_r, \tag{26d}$$

$$T_r = \Gamma - \bar{\lambda}_r \bullet \eta_r \bullet \lambda_r, \tag{26e}$$

$$\Gamma = \varphi^{U\mathrm{irr}} + \sum_r \bar{\lambda}_r \bullet \eta_r \bullet \lambda_r - 2U \tag{26f}$$

$$\varphi^{U\mathrm{irr}} = \Lambda_{2\mathrm{PI}} - U + \sum_r M_r, \tag{26g}$$

$$M_r = (T_r - M_r) \circ \Pi_r \circ T_r = T_r \circ \Pi_r \circ (T_r - M_r) \tag{26h}$$

$$-\Sigma = (\eta_p \bullet \lambda_p) \cdot G = (\bar{\lambda}_p \bullet \eta_p) \cdot G. \tag{26i}$$

As before, they require only the fully two-particle irreducible vertex $\Lambda_{2\mathrm{PI}}$ as an input. Notably, if one employs the so-called SBE approximation [80], which amounts to setting $\Lambda_{2\mathrm{PI}} = U$ as in the parquet approximation *and* neglecting multi-boson exchange contributions $M_r = 0$, all objects involved depend on at most two frequencies. This scheme is therefore numerically favorable compared to the PA if the SBE approximation can be justified [91] . In the context of this paper, we do not employ the SBE approximation, but include multi-boson exchange (MBE) contributions.

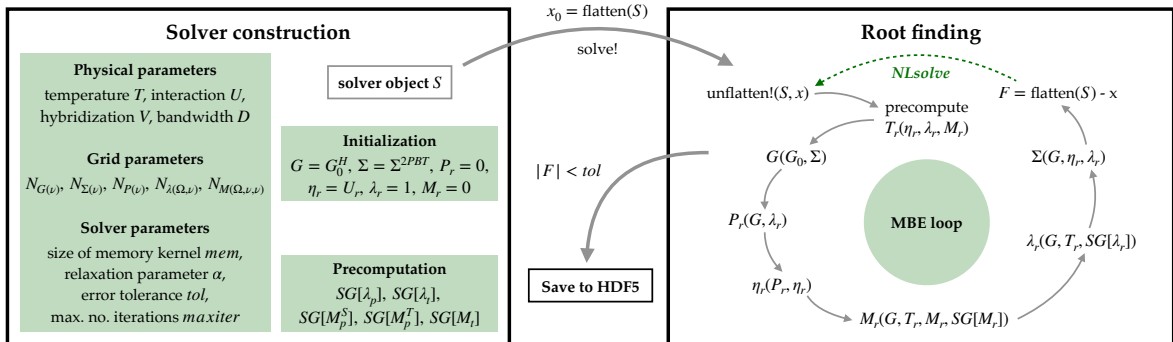

**Figure 5: Structure of the MBE code.** First, an instance $S$ of type `MBEsolver` is constructed by passing the SIAM parameters $T$, $U$, $V$ and $D$ and the sizes for the Matsubara grids. The self-energy $\Sigma$ is initialized using second order perturbation theory (PT2), while all other `MatsubaraFunctions` are set to their bare values. In an optional step, `MatsubaraSymmetryGroups` for $\lambda_r$ and $M_r$ (here denoted by $SG$) can be precomputed. Next, the `solve!` function is used to find the fixed-point of the MBE equations using Anderson acceleration. To interface with `NLsolve`, the fields $\Sigma$, $\eta_r$, $\lambda_r$ and $M_r$ of $S$ (which are sufficient to determine all other involved quantities) are flattened into a single one-dimensional array. After convergence, $S$ is finally written to disk in H5 file format.

### 4.3.2 Implementation in `MatsubaraFunctions.jl`

In this section, we present the implementation of the PA in its MBE formulation using `MatsubaraFunctions.jl`. In doing so, we build upon the code structure developed in Sec. 4.2, i.e. we first define a solver class for which we later implement the self-consistent equations, as well as an interface to solve for the fixed point using Anderson acceleration, see Fig. 5. In order to keep the discussion concise, we refrain from showing all of the code and, instead, focus on computational bottlenecks and point out tricks to circumvent them. For completeness, however, we also make the entire code available via an open-source repository on Github, see Ref. [62] and provide additional implementation details in App. B.

Extending the `GWsolver` from Sec. 4.2 to the `MBEsolver` needed here is a straightforward endeavor, since we just have to add containers and symmetry groups for the Hedin and multi-boson vertices. Furthermore, we extend the solver to include buffers which store the result of evaluating Eqs. (26)(c), (d) and (h), such that repetitive allocations of the multidimensional data arrays for $\lambda_r$ and $M_r$ are avoided. Note that, due to the symmetries of the SIAM studied here, it suffices to include either $\lambda_r$ or $\bar{\lambda}_r$, since $\lambda_r = \bar{\lambda}_r$. In addition, all containers can be implemented as real-valued[16].

```
1  function calc_T(
2      w    :: MatsubaraFrequency,
3      v    :: MatsubaraFrequency,
4      vp   :: MatsubaraFrequency,
5      η_S :: MatsubaraFunction{1, 1, 2, Float64},
6      λ_S :: MatsubaraFunction{2, 1, 3, Float64},
7      η_D :: MatsubaraFunction{1, 1, 2, Float64},
8      λ_D :: MatsubaraFunction{2, 1, 3, Float64},
9      η_M :: MatsubaraFunction{1, 1, 2, Float64},
10     λ_M :: MatsubaraFunction{2, 1, 3, Float64},
11     M_S :: MatsubaraFunction{3, 1, 4, Float64},
```

---

[16]The Green's function $G$ and the self-energy $\Sigma$ are purely imaginary, such that $G = -i\tilde{G}$ and $\Sigma = -i\tilde{\Sigma}$. After plugging this factorization into Eqs. (26), all factors of $i$ are cancelled out such that the resulting equations are entirely real.

```
12      M_T :: MatsubaraFunction{3, 1, 4, Float64},
13      M_D :: MatsubaraFunction{3, 1, 4, Float64},
14      M_M :: MatsubaraFunction{3, 1, 4, Float64},
15      U   :: Float64,
16          :: Type{ch_D}
17      )   :: Float64
18
19      # bare contribution
20      T = -2.0 * U
21
22      # SBE contributions
23      w1       = w + v + vp
24      η1_idx  = MatsubaraFunctions.grid_index_extrp(w1, grids(η_D, 1))
25      λ1_idx1 = MatsubaraFunctions.grid_index_extrp(w1, grids(λ_D, 1))
26      λ1_idx2 = MatsubaraFunctions.grid_index_extrp(vp, grids(λ_D, 2))
27      λ1_idx3 = MatsubaraFunctions.grid_index_extrp( v, grids(λ_D, 2))
28
29      w2       = vp − v
30      v2       = w + v
31      η2_idx  = MatsubaraFunctions.grid_index_extrp(w2, grids(η_D, 1))
32      λ2_idx1 = MatsubaraFunctions.grid_index_extrp(w2, grids(λ_D, 1))
33      λ2_idx2 = MatsubaraFunctions.grid_index_extrp(v2, grids(λ_D, 2))
34
35      T += +0.5 * λ_S[λ1_idx1, λ1_idx2, 1] * η_S[η1_idx, 1] * λ_S[λ1_idx1, λ1_idx3, 1]
36      T += -0.5 * λ_D[λ2_idx1, λ1_idx3, 1] * η_D[η2_idx, 1] * λ_D[λ2_idx1, λ2_idx2, 1]
37      T += -1.5 * λ_M[λ2_idx1, λ1_idx3, 1] * η_M[η2_idx, 1] * λ_M[λ2_idx1, λ2_idx2, 1]
38
39      # MBE contributions
40      w_idx  = MatsubaraFunctions.grid_index_extrp( w, grids(M_S, 1))
41      v_idx  = MatsubaraFunctions.grid_index_extrp( v, grids(M_S, 2))
42      vp_idx = MatsubaraFunctions.grid_index_extrp(vp, grids(M_S, 2))
43
44      w1_idx = MatsubaraFunctions.grid_index_extrp(w1, grids(M_S, 1))
45      w2_idx = MatsubaraFunctions.grid_index_extrp(w2, grids(M_S, 1))
46      v2_idx = MatsubaraFunctions.grid_index_extrp(v2, grids(M_S, 2))
47
48      T += M_D[w_idx, v_idx, vp_idx, 1]
49      T += +0.5 * M_S[w1_idx, v_idx, vp_idx, 1]
50      T += +1.5 * M_T[w1_idx, v_idx, vp_idx, 1]
51      T += -0.5 * M_D[w2_idx, v_idx, v2_idx, 1]
52      T += -1.5 * M_M[w2_idx, v_idx, v2_idx, 1]
53
54      return T
55  end
```

Profiling the MBE code reveals that most of the time is spent calculating the irreducible vertices $T_r$, which are needed to compute both $\lambda_r$ and $M_r$. In the former case, two legs of $T_r$ are closed with a propagator bubble, while in the latter case, $T_r$ enters both to the left and to the right of the respective (Bethe-Salpeter-like) equation. When optimizing the code, it is therefore crucial to find an efficient way to evaluate Eq. (26)(e). In the example above, an exemplary implementation of $T_r$ in the density channel is shown. Here, we make use of the `grid_index_extrp` function, which given a Matsubara frequency and a grid $g$ finds the linear index of the frequency in $g$ or, if it is out of bounds, determines the bound of $g$ that is closest. This function is normally used internally to perform constant extrapolation for `MatsubaraFunction` objects with grid dimension greater than one[17]. Here, however, it can be used to precompute multiple linear indices at once, allowing us to exclusively use the `[]` operator and thus avoid unnecessary boundary checks. Note that we could have used tailfits

---

[17]Therefore it is not exported into the global namespace.

for the screened interactions $\eta_r$ but opt to utilize constant extrapolation instead[18].

Furthermore, when $T_r$ is inserted into the equations for the Hedin and multiboson vertices, it is summed up along one fermionic axis. Therefore, some frequencies, e.g. `w1 = w + v + vp` in line 23 of the code snippet above, will assume the same value for many different external arguments. Hence, to circumvent repeated (but superfluous) `grid_index_extrp` calls, it is beneficial to precompute $T_r$ on a finite grid, which needs to be large enough to maintain the desired accuracy. To this end, we add buffers for the irreducible vertices to our solver class, such that we can compute e.g. the density $T^D$ and magnetic contributions $T^M$ inplace and in parallel, as shown in the example below.

```julia
 1 function calc_T_ph!(
 2     T_D :: MatsubaraFunction{3, 1, 4, Float64},
 3     T_M :: MatsubaraFunction{3, 1, 4, Float64},
 4     η_S :: MatsubaraFunction{1, 1, 2, Float64},
 5     λ_S :: MatsubaraFunction{2, 1, 3, Float64},
 6     η_D :: MatsubaraFunction{1, 1, 2, Float64},
 7     λ_D :: MatsubaraFunction{2, 1, 3, Float64},
 8     η_M :: MatsubaraFunction{1, 1, 2, Float64},
 9     λ_M :: MatsubaraFunction{2, 1, 3, Float64},
10     M_S :: MatsubaraFunction{3, 1, 4, Float64},
11     M_T :: MatsubaraFunction{3, 1, 4, Float64},
12     M_D :: MatsubaraFunction{3, 1, 4, Float64},
13     M_M :: MatsubaraFunction{3, 1, 4, Float64},
14     U   :: Float64
15     )   :: Nothing
16
17     Threads.@threads for vp in grids(T_D, 3)
18         λ1_idx2 = MatsubaraFunctions.grid_index_extrp(vp, grids(λ_D, 2))
19         vp_idx  = MatsubaraFunctions.grid_index_extrp(vp, grids(M_S, 2))
20
21         for v in grids(T_D, 2)
22             w2      = vp - v
23             λ1_idx3 = MatsubaraFunctions.grid_index_extrp( v, grids(λ_D, 2))
24             η2_idx  = MatsubaraFunctions.grid_index_extrp(w2, grids(η_D, 1))
25             λ2_idx1 = MatsubaraFunctions.grid_index_extrp(w2, grids(λ_D, 1))
26             v_idx   = MatsubaraFunctions.grid_index_extrp( v, grids(M_S, 2))
27             w2_idx  = MatsubaraFunctions.grid_index_extrp(w2, grids(M_S, 1))
28
29             for w in grids(T_D, 1)
30                 w1       = w + v + vp
31                 v2       = w + v
32                 η1_idx   = MatsubaraFunctions.grid_index_extrp(w1, grids(η_D, 1))
33                 λ1_idx1  = MatsubaraFunctions.grid_index_extrp(w1, grids(λ_D, 1))
34                 λ2_idx2  = MatsubaraFunctions.grid_index_extrp(v2, grids(λ_D, 2))
35                 w_idx    = MatsubaraFunctions.grid_index_extrp( w, grids(M_S, 1))
36                 w1_idx   = MatsubaraFunctions.grid_index_extrp(w1, grids(M_S, 1))
37                 v2_idx   = MatsubaraFunctions.grid_index_extrp(v2, grids(M_S, 2))
38
39                 # compute SBE vertices
40                 p1 = λ_S[λ1_idx1, λ1_idx2, 1] * η_S[η1_idx, 1] * λ_S[λ1_idx1, λ1_idx3, 1]
41                 p2 = λ_D[λ2_idx1, λ1_idx3, 1] * η_D[η2_idx, 1] * λ_D[λ2_idx1, λ2_idx2, 1]
42                 p3 = λ_M[λ2_idx1, λ1_idx3, 1] * η_M[η2_idx, 1] * λ_M[λ2_idx1, λ2_idx2, 1]
43
44                 # compute MBE vertices
45                 m1 = M_S[w1_idx, v_idx, vp_idx, 1]
46                 m2 = M_T[w1_idx, v_idx, vp_idx, 1]
47                 m3 = M_D[w2_idx, v_idx, v2_idx, 1]
48                 m4 = M_M[w2_idx, v_idx, v2_idx, 1]
49
```

---

[18]Since $\eta_r$ depends only on one frequency argument, it can be stored on a rather large grid, such that its asymptotic behavior is well-captured even without polynomial extrapolation.

```
50                    T_D[w, v, vp] = -2.0 * U + M_D[w_idx, v_idx, vp_idx, 1] + 0.5 * (p1 + m1 - p2 - m3) +
                      ↪ 1.5 * (m2 - p3 - m4)
51                    T_M[w, v, vp] = +2.0 * U + M_M[w_idx, v_idx, vp_idx, 1] - 0.5 * (p1 + m1 + p2 + m3) +
                      ↪ 0.5 * (m2 + p3 + m4)
52                end
53            end
54      end
55
56      return nothing
57 end
```

Here, we also make use of the fact that many frequency arguments (and their respective linear indices) are shared between different channels, which speeds up the calculation of $T$ even further. The implementation of, say, Eq. (26)(h) is now rather straightforward. $M^D$, for example, can be computed as shown below.

```
1 function calc_M!(
2      M    :: MatsubaraFunction{3, 1, 4, Float64},
3      Pi   :: MatsubaraFunction{2, 1, 3, Float64},
4      T    :: MatsubaraFunction{3, 1, 4, Float64},
5      M_D  :: MatsubaraFunction{3, 1, 4, Float64},
6      SG   :: MatsubaraSymmetryGroup,
7           :: Type{ch_D}
8      )    :: Nothing
9
10      # model the diagram
11      function f(wtpl, xtpl)
12
13          w, v, vp  = wtpl
14          val       = 0.0
15          v1, v2    = grids(Pi, 2)(grids(T, 3)[1]), grids(Pi, 2)(grids(T, 3)[end])
16          Pi_slice  = view(Pi, w, v1 : v2)
17          M_D_slice = view(M_D, w, v, :)
18          T_L_slice = view(T, w, v, :)
19          T_R_slice = view(T, w, vp, :)
20
21          vl = grids(T, 3)(grids(M_D, 3)[1])
22          vr = grids(T, 3)(grids(M_D, 3)[end])
23
24          for i in 1 : vl - 1
25              val -= (T_L_slice[i] - M_D_slice[1]) * Pi_slice[i] * T_R_slice[i]
26          end
27
28          for i in vl : vr
29              val -= (T_L_slice[i] - M_D_slice[i - vl + 1]) * Pi_slice[i] * T_R_slice[i]
30          end
31
32          for i in vr + 1 : length(T_L_slice)
33              val -= (T_L_slice[i] - M_D_slice[vr - vl + 1]) * Pi_slice[i] * T_R_slice[i]
34          end
35
36          return temperature(M) * val
37      end
38
39      # compute multiboson vertex
40      SG(M, MatsubaraInitFunction{3, 1, Float64}(f); mode = :hybrid)
41
42      return nothing
43 end
```

Here, we utilize the corresponding `MatsubaraSymmetryGroup` object with the hybrid MPI + `Threads` parallelization scheme. In addition, we make use of views for the bubble and vertices

to avoid repeated memory lookups in the Matsubara summation.

### 4.3.3 Benchmark results

In this section, we benchmark the presented implementation of the MBE parquet solver against an independent implementation in C++. Our motivation for this comparison is twofold: Firstly, we want to verify the overall correctness of both implementations and, secondly, we want to test how robust the multiboson formalism is to implementation details. This regards, for example, the treatment of correlation functions at the boundaries of their respective frequency grids. While the Julia code relies on (polynomial or constant) extrapolation, the C++ code replaces correlators with their asymptotic value instead. Ideally, these details should be irrelevant, except in the most difficult parameter regimes. Both codes used the physical parameters as stated after Eq. (18) and the frequency parameters according to Tab. 1. We begin by examining the static properties of the model including the quasiparticle residue $Z$ given by

$$Z^{-1} = 1 - \frac{\mathrm{dIm}[\Sigma(\omega)]}{\mathrm{d}\omega}\bigg|_{\omega \to 0}, \tag{27}$$

as well as the susceptibilities in the density $(D)$ and magnetic $(M)$ channels. The latter can be obtained from the screened interactions analogous to Ref. [92], that is

$$\chi^{D/M} = \frac{\eta^{D/M} - U^{D/M}}{(U^{D/M})^2}. \tag{28}$$

The corresponding results are summarized in Fig. 6. Both codes are in quantitative agreement and predict a strong enhancement of magnetic fluctuations at low temperatures. However, as has been noted in Ref. [77], the characteristic signature for the formation of a local magnetic moment at the impurity, a decrease of $\chi^D$ for temperatures $T \lesssim 2$ (for the specific choice of numerical parameters used here), is not captured by the parquet approximation. Instead, $\chi^D$ increases monotonically over the entire range of temperatures considered and the system remains in a metallic state with well-defined quasiparticles (i.e. $0 < Z < 1$).

Figure 7 shows a direct comparison of the MBE vertices and their evolution with decreasing temperature within both codes. As can be seen from the middle column, showing the screened interaction, Hedin and multiboson vertex in the magnetic channel, most of the long-lived magnetic correlations are already captured by the screened interaction itself and thus by the corresponding single-boson exchange diagrams. In contrast, low-energy scattering processes mediated by multiple bosons seem to be less relevant, as indicated by a comparatively small $M^M$ contribution. This picture is somewhat reversed in the other channels (left and right column in Fig. 7). In the density channel, for example, the largest contribution originates from short and also long-lived multiboson fluctuations, especially at low temperatures.

Figure 8 presents further results for $M^X$ as a function of its two fermionic frequencies $\nu$ and $\nu'$ (with fixed $\Omega = 0$). Remarkably, the structure of these objects is dominated by cross-like structures similar to those discussed in Ref. [92], which become more pronounced when $T$ is decreased. A comparison of the data obtained with both codes (shown in the second row of Fig. 8), reveals that it is precisely these structures that seem difficult to capture in numerical calculations, and where small differences in the implementation can have a significant effect. However, the relative difference between the results from both codes is still small ($\lesssim 0.01$).

|     | total no. frequencies |
| --- | --- |
| $G$ | 4096 |
| $\Sigma$ | 512 |
| $\eta$ | 1023 |
| $\lambda$ | $575 \times 512$ |
| $M$ | $383 \times 320 \times 320$ |

**Table 1: Frequency parameters for the benchmark results in Figs. 6-9.** We show the total number of frequencies used for the various Matsubara functions. Since the boxes are symmetric around zero there is an even (odd) number of Matsubara frequencies along fermionic (bosonic) directions.

As a final benchmark of the codes, we have considered their respective serial and parallel performance for a single evaluation of the parquet equations in SBE decomposition (see Fig. 9). Surprisingly, the Julia code based on `MatsubaraFunctions.jl` outperforms the C++ implementation by about a factor of four when run in production mode (i.e., with internal sanity checks disabled). We would like to note that this is most likely *not* due to a fundamental performance advantage of Julia over C++, but simply the result of several optimizations (such as those presented in Sec. 4.3.2) that were more easy to implement using `MatsubaraFunctions.jl`.

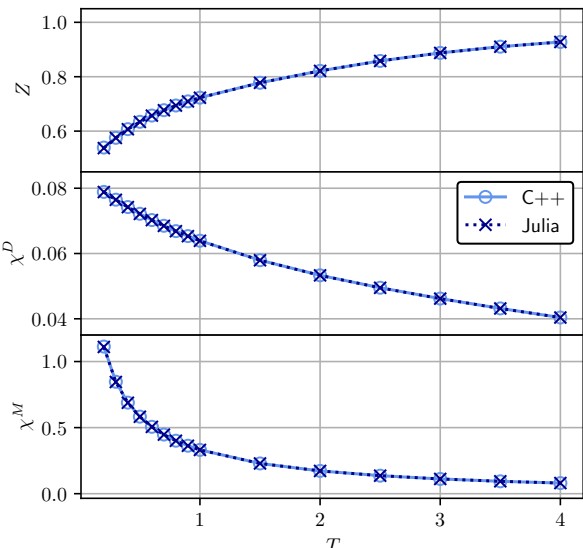

**Figure 6: Results for the quasi-particle residue $Z$ and the density/magnetic susceptibility $\chi^{D/M}$.** The comparison shows good agreement between the two codes. Note that we approximated the derivative in Eq. (27) by a fourth order finite differences method.

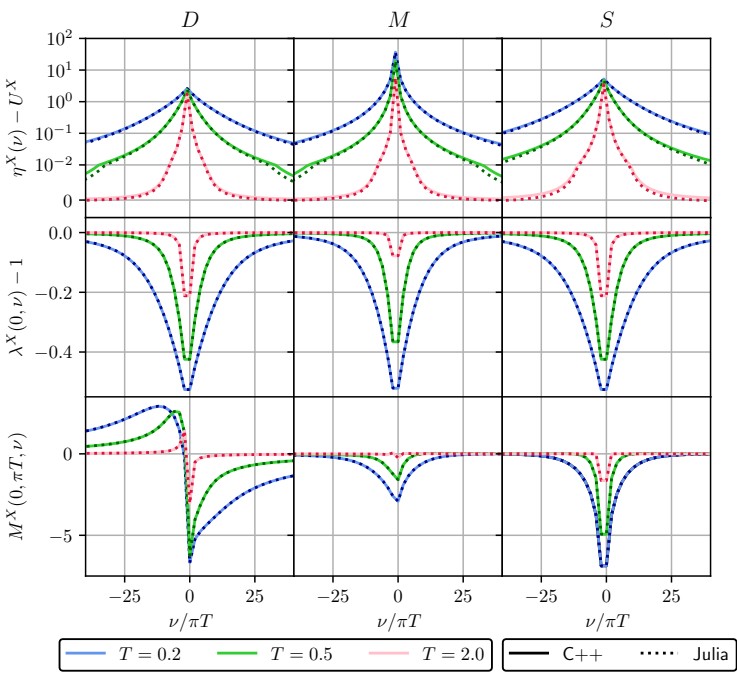

**Figure 7: Benchmark of vertex quantities between the Julia and C++ code.** We show frequency slices through various SBE ingredients (top to bottom: screened interactions, Hedin vertices, multiboson vertices) at different temperatures and channels. The comparison shows good agreement between both codes.

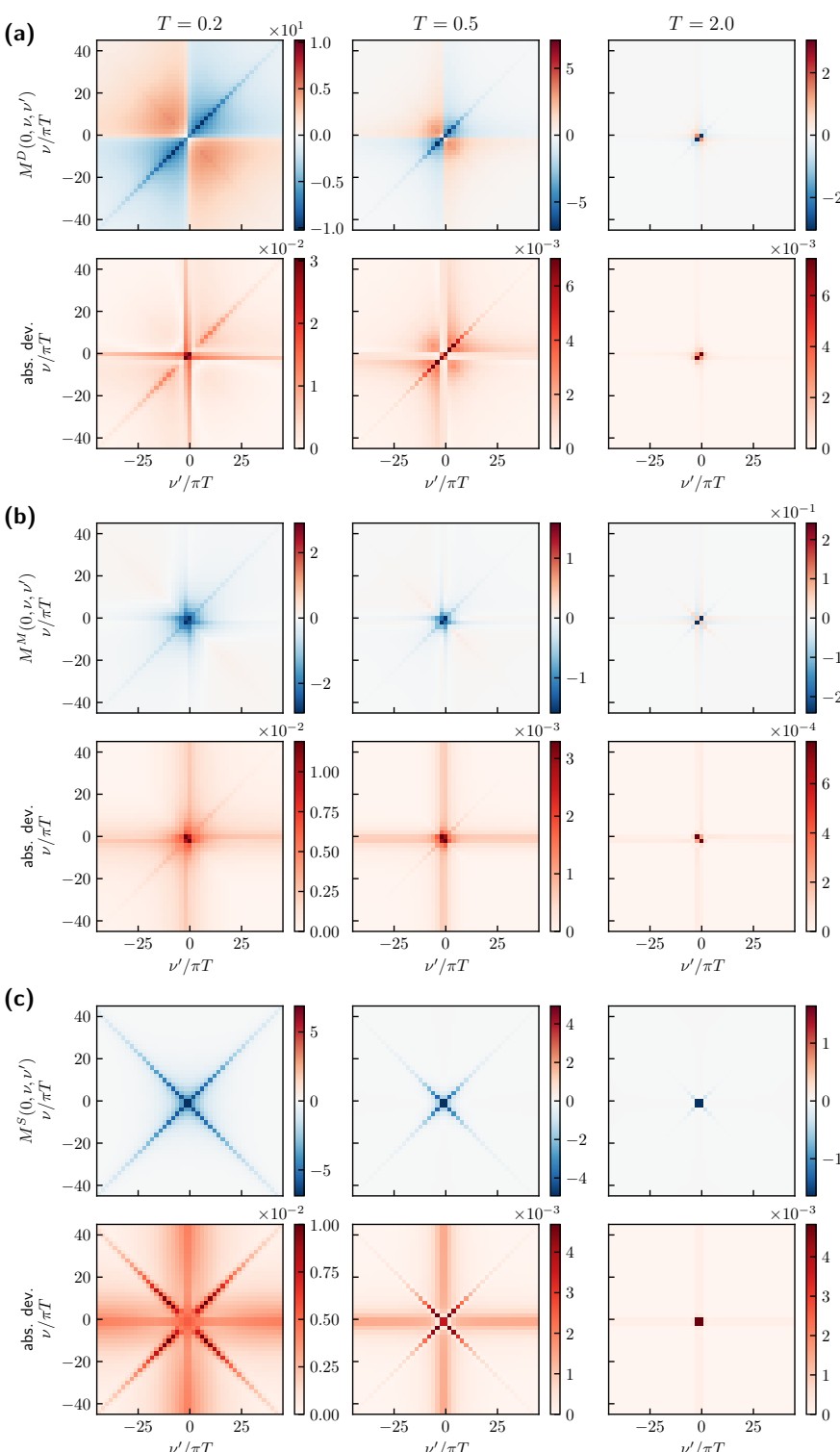

**Figure 8: Slice through multi-boson contributions $M^D$, $M^M$ and $M^S$.** The upper panels show the data for different temperatures, the lower panels the absolute deviation between the Julia and the C++ implementation, respectively. For lower temperatures the features in the data require the computation and storage of a larger number of frequency points. The agreement of the data persists to the lowest temperature shown in this paper.

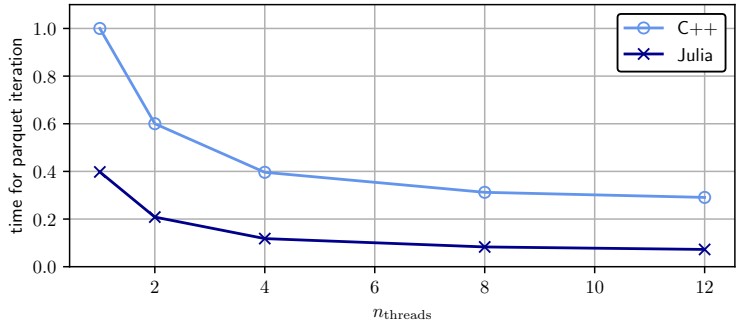

**Figure 9: Performance benchmark between the Julia and C++ code.** We show the time taken for a single evaluation of the parquet equations in SBE decomposition. Note that the runtimes have been normalized to the serial result of the C++ code.

## 5   Future directions

We have presented a first version of the `MatsubaraFunctions.jl` library and its basic functionality. Although the library already offers many features, most notably an automated interface for implementing and exploiting symmetries when working with Green's functions (including several options for parallel evaluation), as well as high performance for larger projects (see Sec. 4.3.1 and the discussions therein), several generalizations of the interface and further optimizations are currently under development. In addition, we will add more support for generic grid types other than just Matsubara frequency grids. These could include, for example, momentum space grids and support for continuous variables (such as real frequencies). Note, however, that calculations in momentum or real space are already feasible with the current state of the package, if a suitable mapping from, say, wavevectors to indices is provided. Accuracy improvements for fitting high-frequency tails and more advanced extrapolation schemes for Matsubara sums are also in the works.

In the future, it will be very valuable to extend the ecosystem surrounding `MatsubaraFunctions.jl`. For example, many state-of-the-art diagrammatic solvers rely on the efficient evaluation of similar diagrams such as vertex-bubble-vertex contractions, which are a common feature of Bethe-Salpeter-type equations. These operations could be developed independently of the main library, providing even more quality-of-life options for the user. Moreover, such a toolkit would allow for the swift deployment of different types of solvers, including fRG solvers for quantum spin systems and self-consistent impurity solvers such as the MBE code presented in Sec. 4.3.2, to name but a few. With many new and exciting correlated materials becoming available, fast and flexible solvers are of utmost importance to facilitate scientific progress, and we strongly believe that a package like `MatsubaraFunctions.jl` could be a useful tool for their rapid development.

## 6   Acknowledgements

We would like to thank Fabian Kugler, Jae-Mo Lihm, Seung-Sup Lee, Friedrich Krien, Marc Ritter, Björn Sbierski and Benedikt Schneider for helpful discussions and collaboration on ongoing projects. This work was funded in part by the Deutsche Forschungsgemeinschaft

under Germany's Excellence Strategy EXC-2111 (Project No. 390814868). It is part of the Munich Quantum Valley, supported by the Bavarian state government with funds from the Hightech Agenda Bayern Plus. N.R. acknowledges funding from a graduate scholarship from the German Academic Scholarship Foundation (Studienstiftung des deutschen Volkes) and additional support from the "Marianne-Plehn-Programm" of the state of Bavaria. The numerical simulations were, in part, performed on the Linux clusters and the SuperMUC cluster (project 23769) at the Leibniz Supercomputing Center in Munich. The Flatiron Institute is a division of the Simons Foundation.

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

# A  Extrapolation of Matsubara sums

Suppose we want to compute the fermionic Matsubara sum $f(\tau \to 0^+) = \frac{1}{\beta} \sum_\nu f(\nu) e^{-i\nu 0^+}$. We assume that $f(z)$ with $z \in \mathbb{C}$ has a Laurent series representation in an elongated annulus about the imaginary axis which decays to zero for large $|z|$. If the poles and residues of $f$ in the complex plane are known, this problem can in principle be solved by rewriting the Matsubara sum as a contour integral and applying Cauchy's residue theorem after deforming the contour. Unfortunately, these poles are usually unknown and we have to resort to numerical calculations instead. There, however, we can only compute the sum over a finite (symmetric) grid of Matsubara frequencies, which converges very slowly if at all.

To tackle this problem, let us assume that $f$ is known on a grid with sufficiently large maximum (minimum) frequency $\pm\Omega$, such that we can approximate

$$f(\nu) \approx \sum_{n=1}^{N} \frac{\alpha_n}{(i\nu)^n}, \tag{29}$$

for $|\nu| > \Omega$. Neglecting the factor $e^{-i\nu 0^+}$ for brevity, this allows us to split up the expression for $f(\tau \to 0^+)$ as

$$\frac{1}{\beta} \sum_{\nu} f(\nu) = \frac{1}{\beta} \sum_{\nu < -\Omega} f(\nu) + \frac{1}{\beta} \sum_{-\Omega \leq \nu \leq \Omega} f(\nu) + \frac{1}{\beta} \sum_{\nu > \Omega} f(\nu)$$

$$\approx \frac{1}{\beta} \sum_{\nu < -\Omega} \sum_{n=1}^{N} \frac{\alpha_n}{(i\nu)^n} + \frac{1}{\beta} \sum_{-\Omega \leq \nu \leq \Omega} f(\nu) + \frac{1}{\beta} \sum_{\nu > \Omega} \sum_{n=1}^{N} \frac{\alpha_n}{(i\nu)^n}, \tag{30}$$

where (29) was used to approximate the semi-infinite sums. In many cases, the dominant asymptotic behavior of single-particle Green's functions and one-dimensional slices through higher-order vertex functions is already well captured by an algebraic decay $(i\nu)^{-q}$ with $q = 1, 2$. Therefore, by truncating the asymptotic expansion at $N = 2$, we can rewrite the right-hand side as

$$\frac{1}{\beta} \sum_{\nu} f(\nu) \approx \frac{1}{\beta} \sum_{-\Omega \leq \nu \leq \Omega} f(\nu) + \sum_{n=1}^{2} \left[ \frac{1}{\beta} \sum_{\nu} \frac{\alpha_n}{(i\nu)^n} - \frac{1}{\beta} \sum_{-\Omega \leq \nu \leq \Omega} \frac{\alpha_n}{(i\nu)^n} \right]. \tag{31}$$

The series in the bracket can be computed straightforwardly using Cauchy's residue theorem and we find

$$\frac{1}{\beta} \sum_{\nu} f(\nu) e^{-i\nu 0^+} \approx \frac{1}{\beta} \sum_{-\Omega \leq \nu \leq \Omega} \left[ f(\nu) - \frac{\alpha_2}{(i\nu)^2} \right] - \frac{\alpha_1}{2} - \beta \frac{\alpha_2}{4}. \tag{32}$$

Thus, if the coefficients $\alpha_n$ are known (for example by fitting the high-frequency tails), this formula can provide a quick and dirty approximation to the infinite Matsubara sum.

## B    Implementation details for the MBE solver

In this section we provide additional information on the implementation of the MBE equations, which were introduced on a general basis in Sec. 4.3.1 of the main text. As for any application involving many-body Green's functions, it is crucial to choose an appropriate parametrization of the self-consistent equations that reflects the symmetries of the field theory under consideration. Here, we deal with the implementation of SU(2) symmetry (spin rotation invariance) as well as time translation invariance (energy conservation) for the MBE equations of the impurity model defined in Sec. 4.3.

### B.1    SU(2) symmetry

Consider an SU(2) transformation $U = e^{i\boldsymbol{\epsilon}\boldsymbol{\sigma}}$, where $\boldsymbol{\epsilon} \in \mathbb{R}^3$ and $\boldsymbol{\sigma}$ is the vector of Pauli matrices. Under $U$, the fermionic creation and annihilation operators transform into

$$c_s \to U_{ss'} c_{s'}, \ c_s^\dagger \to c_{s'}^\dagger (U^\dagger)_{s's}, \tag{33}$$

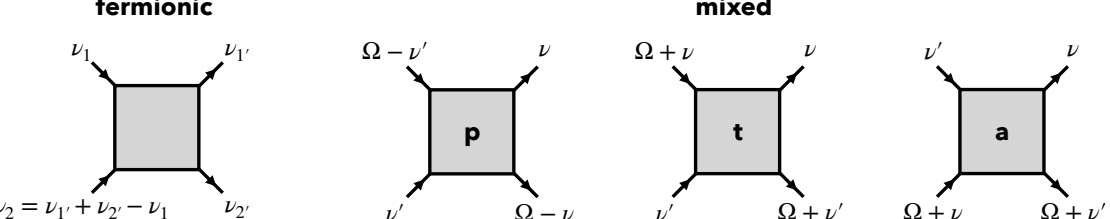

**Figure 10: Mixed frequency conventions.** In mixed notation, each 2PR channel is described in terms of one bosonic argument $\Omega$ and two fermionic frequencies $\nu, \nu'$ as opposed to the purely fermionic notation shown on the left.

where we have omitted all indices except the spin $s = \{\uparrow, \downarrow\}$. For SU(2) symmetric actions it can be shown that single-particle Green's functions $G^{(1)}_{ss'}$ are diagonal and also invariant under spin flips, i.e. $G^{(1)}_{ss'} = G^{(1)}\delta_{ss'}$ [48]. Two-particle correlators $G^{(2)}_{s_{1'}s_1s_{2'}s_2}$, on the other hand, can be decomposed into two components $G^{(2)|=}$ and $G^{(2)|\times}$, which preserve the total spin between incoming and outgoing particles

$$G^{(2)}_{s_{1'}s_1s_{2'}s_2} = G^{(2)|=}\delta_{s_{1'}s_1}\delta_{s_{2'}s_2} + G^{(2)|\times}\delta_{s_{1'}s_2}\delta_{s_{2'}s_1} \,. \tag{34}$$

Furthermore, the Bethe-Salpeter-like equations (24) can be diagonalized by introducing a singlet ($S$) and a triplet ($T$) component

$$
\begin{aligned}
G_p^{(2)|S} &= G_p^{(2)|=} - G_p^{(2)|\times} \\
G_p^{(2)|T} &= G_p^{(2)|=} + G_p^{(2)|\times} \,,
\end{aligned}
\tag{35}
$$

in the $p$ channel, and a density ($D$) and magnetic ($M$) contribution

$$
\begin{aligned}
G_t^{(2)|D} &= 2G_t^{(2)|=} + G_t^{(2)|\times} \\
G_t^{(2)|M} &= G_t^{(2)|\times} \,,
\end{aligned}
\tag{36}
$$

in the $t$ channel. Moreover, this change of basis has the advantage that physical response functions can be obtained directly from the screened interaction in the respective channel. The spin susceptibility $\chi^M$, for example, is simply given by $-U^2\chi^M = \eta^M + U$ for a local Hubbard $U$. For this reason, the $\{S, T, D, M\}$ basis is sometimes called the *physical* spin basis, whereas the decomposition into parallel ($=$) and crossed terms ($\times$) is known as the *diagrammatic* spin basis [48]. In the implementation of the MBE solver, the former is used.

## B.2 Time translation invariance

The interacting part of the impurity action from Sec. 4.3 is static, i.e. the bare interaction $U$ is $\tau$-independent. Consequently, one and two-particle Green's functions are invariant under translations in imaginary time, which implies conservation of the total Matsubara frequency between incoming and outgoing legs [48] and, thus,

$$
\begin{aligned}
G^{(1)}(\nu, \nu') &= G^{(1)}(\nu) \times \beta\delta_{\nu|\nu'} \\
G^{(2)}(\nu_{1'}, \nu_1, \nu_{2'}, \nu_2) &= G^{(2)}(\nu_{1'}, \nu_1, \nu_{2'}) \times \beta\delta_{\nu_{1'}+\nu_{2'}|\nu_1+\nu_2} \,.
\end{aligned}
\tag{37}
$$

Note that we have suppressed additional indices, such as spin, for brevity. For two-particle quantities, it is convenient to adopt a *mixed* frequency convention for the 2PR channels, where, instead of three fermionic arguments, one bosonic transfer frequency $\Omega$ and two fermionic frequencies $\nu, \nu'$ are used. The convention used for the MBE solver is shown in Fig. 10.