# Peer review of "MatsubaraFunctions.jl: An equilibrium Green's function library in the Julia programming language"

_SciPost Physics Codebases, doi:SciPost Phys. Codebases 24-r0.1 (2024) , SciPost Phys. Codebases 24 (2024)_

## Round 1 · Referee Report · Anonymous (Referee 1) · 2023-11-8

Strengths

  1. Usefulness of the library (for julia programmers)
  2. Performance is impressive
  3. Real-space continuation is possible (via Padé approximants)

Weaknesses

  1. The documentation seems limited to this paper and is not systematic
  2. The discussion about symmetries in the paper is rather sketchy.
  3. Not clear this library can deal with wavevectors

Report

Let me expand on the above strengths and weaknesses.

First the strengths.
Functions of one or more Matsubara frequencies occur regularly in computations involving Green functions in many-body physics and this library will do great service to people engaged in this type of calculations, provided of course that they are willing to use the julia language
Performance is impressive, and compares favorably, in the case of the parquet problem, with a different solver written in C++ (but less optimized, according to the authors).
Real-space continuation is possible via Padé approximants, which however requires extended precision.

Then weaknesses.
The main weakness in my view is that the documentation seems limited to this paper (which does not thoroughly describe all features of the library) and (I presume) the online help available in julia itself. A full documentation (e.g. in Sphinx) would be necessary, I think. if such a documentation exists, it should be pointed out ou rendered public on github. If not, it should be put in place, as this is not so hard.

The discussion about symmetries in the paper is rather sketchy. I haven't seen a proper definition of the structures MatsubaraSymmetry and MatsubaraSymmetryGroup. Only examples are given. Again this goes back to the question of complete documentation.

It is not clear that this library can deal with wavevectors, not just lattice sites. In other words, in Eq. (1), could the index i_1 be a wavevector as well? I suppose so, but the atomic limit taken in the first two examples raises a doubt. The parquet example seems agnostic on this matter. I suppose the main difference is that a wavevector grid could be nonuniform and in that way better represent the low-energy physics. Comments on this matter would be welcome.

Requested changes

minor typo: monilithic --> monolithic

  • validity: high
  • significance: high
  • originality: high
  • clarity: good
  • formatting: excellent
  • grammar: excellent

Author:  Dominik Kiese  on 2023-11-29  [id 4160]

(in reply to Report 1 on 2023-11-08)

We thank the referee for being in favor of the publication of our paper. It seems that there are two questions raised, one regarding documentation, the other one regarding implementation of momentum space grids. We will try to answer them one by one.

(1) We agree with the referee that a full documentation of public code is essential to facilitate its usage beyond a small community of developers. We want to comment, however, that our code is indeed fully documented. To reach the documentation, users can simply click on the "Documentation"-badge at the top of our github repository. This will refer them to another webpage where all functions exported by our package are listed with their complete type signature and short explanations, including a search function to browse the documentation more quickly. We have also included the examples from our paper in there to give new users, which may not be aware of the paper, an overview of its features. To make this point more clear, we reference the documentation more explicitly at the beginning of Section 3:
"A full documentation of the package is available from the github repository."

(2) The referee makes an important point here. In the paper, we have indeed focussed on the case where the indices i1, ..., iN are associated with lattice sites or orbitals. In general, however, such a distinction is not necessary for the data structures we define in our package. If someone attempts to use the library for calculations in momentum space, this can indeed be done if a mapping from wavevectors to indices is provided by the user. The MatsubaraFunction struct is then of course agnostic to the details of the k-grid, e.g. whether it is uniform or non-uniform. In a future release we plan to ship such index mappings directly with the library as a quality-of-life feature. We comment on this circumstance in Sec. 5:
"Note, however, that calculations in momentum or real space are already feasible with the current state of the package, if a suitable mapping from, say, wavevectors to indices is provided."

---

## Round 1 · Referee Report · Anonymous (Referee 2) · 2023-11-16

Strengths

(1) Useful for many numerical applications in solid state physics
(2) Presents good educational example applications
(3) High performance

Weaknesses

(1) Comparison between Julia and C++ codes not clear
(2) Fig. 6 doesn't show difference between Julia and C++ codes
(3) Origin of high performance could be better discussed

Report

This manuscript introduces a Julia library for handling n-point Matsubara Green's functions in numerical quantum many-particle applications. This includes the definition of data structures, the evaluation of Matsubara sums, symmetry reduction, parallelization and more. The usage is demonstrated in code snippets and in two concrete physics applications. These applications focus on a single site Hubbard model in Hartree-Fock and GW approximations as well as on the more involved single impurity Anderson model in "multi-boson exchange" formalism.

In total, the work is very well motivated and nicely presented. I agree with the authors that there are plenty of numerical quantum many-body applications where the developed library can be useful. Furthermore, the presented applications are educational from a user perspective but at the same time physically non-trivial and interesting. Therefore, I highly recommend the publication of this manuscript.

I only have very few comments:

(1) I do not quite understand what one can learn from the comparison in Sec. 4.3.3 where results from a Julia code and from a C++ code are compared. Are both codes using identical algorithms? Then, I would expect identical results (up to machine precision) and there is no need to compare results. If they use different algorithms, it would be important to discuss these differences. Otherwise, it is unclear how differences in the results should be interpreted.

(2) This comment is related to (1). By naked eye, I cannot see any differences in Fig. 6 between the results from the Julia and C++ codes and, therefore, (in the absence of any further information) I would conclude they are identical. Since the authors say the agreement is "good" I guess this is not the case. Then, a better way of illustrating the differences might be helpful. Similar for Fig. 7; however, there, small differences are visible.

(3) On the bottom of page 24, the authors mention that the Julia code is four times faster than the C++ code. This is a substantial factor that makes a big difference! However, its possible origin is only discussed very briefly. A longer discussion of this substantial effect might be helpful.

(4) A minor error: The link behind Eq. (26) does not lead to Eq. (26) but somewhere else.

Requested changes

Fix the link behind Eq. (26)

  • validity: high
  • significance: high
  • originality: high
  • clarity: good
  • formatting: good
  • grammar: perfect

Author:  Dominik Kiese  on 2023-11-29  [id 4159]

(in reply to Report 2 on 2023-11-16)
Category:
remark
answer to question

We appreciate the referee's recommendation for publication of our manuscript. Here, we want to give a short assessment of the provided comments:

(1) & (2): We thank the referee for giving us the opportunity of clarifying this point. While both codes in principle implement the same algorithm ( in the sense that the diagrammatic resummation is identical), they differ in minor details. For example, the data structures provided by the MatsubaraFunctions library uses constant extrapolation for out-of-bounds access of multi-variate correlation functions (see Section 3.2), whereas the C++ implementation sets the multiboson (and related) vertices to their asymptotic value. The purpose of Section 4.3.3 is twofold: (a) we want to verify the overall correctness of both implementations (regarding signs, prefactors etc.), and (b) we want to test how robust the method is to those implementation details. Ideally, these differences should not matter (except in extreme parameter regimes). This is, for example, illustrated in Figure 8 where we plot the relative error in the multiboson vertices M (the most difficult object to compute) as a function of temperature. It can be seen that discrepancies between the codes remain below the percent level even at the lowest temperatures considered. We have added a short description of this motivation to the beginning of Section 4.3.3:
"Our motivation for this comparison is twofold: Firstly, we want to verify the overall correctness of both implementations and, secondly, we want to test how robust the multiboson formalism is to implementation details. This regards, for example, the treatment of correlation functions at the boundaries of their respective frequency grids. While the Julia code relies on (polynomial or constant) extrapolation, the C++ code replaces correlators with their asymptotic value instead. Ideally, these details should be irrelevant, except in the most difficult parameter regimes."

(3): We thank the referee for pointing this out. The difference in speed mainly results from the code optimizations presented in Section 4.3.2, which, while present in the Julia implementation, are lacking in the C++ code. This section is also devoted to demonstrating that the simplistic API of the library allows one to discover, and, subsequently, remove bottlenecks which might be harder to fix if the code is more convoluted. To clarify on this matter, a reference pointing out the relation to Section 4.3.2 was added to the manuscript:
"We would like to note that this is most likely not due to a fundamental performance advantage of Julia over C++, but simply the result of several optimizations (such as those presented in Sec. 4.3.2) that were more easy to implement using MatsubaraFunctions.jl."

(4): We agree with the referee that problems with hyper-references should be fixed. However, we are not sure to which link the referee refers, since there is no reference to an equation behind Eq. (26) in Section 4.3.1, as far as we can see. If possible, could the referee point us to the exact location where this issue is observed?

---

## Round 2 · Referee Report · Anonymous (Referee 2) · 2023-12-1

Report

I thank the authors for addressing my comments and questions. In my opinion, the paper is now ready for publication.

Concerning the minor issue with Eq. (26): This equation is referenced many times in the text, for example in the footnote on page 20. However, when I click on the link behind the citation of Eq. (26), I don't end up on page 19, where the equations 26 are shown, but on page 12. This occurs for various pdf viewers which I tried.

---

## Round 2 · Referee Report · Anonymous (Referee 1) · 2023-12-17

Report

The authors made the requested changes. I think the paper is not fit for publication.

Requested changes

None

---

## Round 2 · List of Changes

Added to Section 4.3.3: "Our motivation for this comparison is twofold: Firstly, we want to verify the overall correctness of both implementations and, secondly, we want to test how robust the multiboson formalism is to implementation details. This regards, for example, the treatment of correlation functions at the boundaries of their respective frequency grids. While the Julia code relies on (polynomial or constant) extrapolation, the C++ code replaces correlators with their asymptotic value instead. Ideally, these details should be irrelevant, except in the most difficult parameter regimes."

Changed in Section 4.3.3: "We would like to note that this is most likely not due to a fundamental performance advantage of Julia over C++, but simply the result of several optimizations (such as those presented in Sec. 4.3.2) that were more easy to implement using MatsubaraFunctions.jl."

Added to Section 3: "A full documentation of the package is available from the github repository."

Added to Section 5: "Note, however, that calculations in momentum or real space are already feasible with the current state of the package, if a suitable mapping from, say, wavevectors to indices is provided."

---

## Editorial Decision

published